# The iron–sulfur helicase DDX11 promotes the generation of single-stranded DNA for CHK1 activation

Anna K Simon[1], Sandra Kummer[1], Sebastian Wild[1,*], Aleksandra Lezaja[2,*], Federico Teloni[2,* ●], Stanislaw K Jozwiakowski[1 ●], Matthias Altmeyer[2], Kerstin Gari[1 ●]

**The iron–sulfur (FeS) cluster helicase DDX11 is associated with a human disorder termed Warsaw Breakage Syndrome. Interestingly, one disease-associated mutation affects the highly conserved arginine-263 in the FeS cluster-binding motif. Here, we demonstrate that the FeS cluster in DDX11 is required for DNA binding, ATP hydrolysis, and DNA helicase activity, and that arginine-263 affects FeS cluster binding, most likely because of its positive charge. We further show that DDX11 interacts with the replication factors DNA polymerase delta and WDHD1. In vitro, DDX11 can remove DNA obstacles ahead of Pol $\delta$ in an ATPase- and FeS domain-dependent manner, and hence generate single-stranded DNA. Accordingly, depletion of DDX11 causes reduced levels of single-stranded DNA, a reduction of chromatin-bound replication protein A, and impaired CHK1 phosphorylation at serine-345. Taken together, we propose that DDX11 plays a role in dismantling secondary structures during DNA replication, thereby promoting CHK1 activation.**

## Introduction

The DNA helicase DDX11 (also known as ChlR1) is a member of a subfamily of SF2 helicases that share a sequence motif containing four conserved cysteines that can coordinate a $[4Fe-4S]^{2+}$ cluster (Rudolf et al, 2006). In the related helicase xeroderma pigmentosum group D-complementing protein (XPD)/Rad3, the FeS cluster constitutes a structural element that is able to couple ATP hydrolysis to translocation along single-stranded DNA (ssDNA) and is, hence, required for DNA unwinding (Rudolf et al, 2006; Pugh et al, 2008), but the function of the FeS cluster in DDX11 has not been addressed yet.

On the biochemical level, DDX11 is a DNA-dependent ATPase that can unwind DNA:DNA and DNA:RNA duplexes with a 5′–3′ polarity (Hirota & Lahti, 2000). To do so, DDX11 requires a 5′-ssDNA overhang with a minimal length of 15 nucleotides for helicase loading (Wu et al, 2012). DDX11 shows a preference for short forked duplex substrates, but can readily unwind 5′-flap structures, 5′-tailed D-loop substrates, anti-parallel G-quadruplex DNA (Wu et al, 2012), and melt inter- and intra-molecular DNA triplex substrates (Guo et al, 2015).

Biallelic mutations in *DDX11* result in a rare disease termed Warsaw breakage syndrome (WABS) that is associated with severe developmental defects, including microcephaly, growth retardation, and facial dysmorphy (van der Lelij et al, 2010; Capo-Chichi et al, 2013; Bailey et al, 2015; Alkhunaizi et al, 2018). Cells derived from WABS patients display drug-induced chromosomal breakage reminiscent of Fanconi anaemia cells and sister chromatid cohesion defects (van der Lelij et al, 2010).

A role for DDX11 in sister chromatid cohesion establishment and chromosome segregation has been further confirmed using various model systems from yeast to human (Petronczki et al, 2004; Skibbens, 2004; Parish et al, 2006), but the actual function of DDX11 in this process has remained unclear. Moreover, although DDX11 was found to be important for the retention of the cohesin complex on chromatin in yeast and human (Borges et al, 2013; Cortone et al, 2018), there seem to be organism-specific differences with respect to the contribution of its helicase activity, which was found to be dispensable for cohesion establishment in *Saccharomyces cerevisiae* (Samora et al, 2016), whereas being essential in chicken DT-40 cells (Abe et al, 2016). In human cells, an ATPase-dead version of DDX11 could partially restore cohesion establishment upon DDX11 depletion (Cortone et al, 2018), suggesting that in humans, DDX11 may contribute to cohesion establishment in ways that are both helicase-dependent and helicase-independent.

Interestingly, three siblings with WABS have been found to be homozygous carriers of a mutation that causes a single amino acid change that affects a highly conserved arginine residue located within the FeS domain of DDX11 (Capo-Chichi et al, 2013). Biochemically, this arginine-to-glutamine variant (R263Q DDX11) was found to be largely inactive with impaired DNA binding, ATP

[1]Institute of Molecular Cancer Research, University of Zurich, Zurich, Switzerland   [2]Department of Molecular Mechanisms of Disease, University of Zurich, Zurich, Switzerland

Correspondence: gari@imcr.uzh.ch
Federico Teloni's present address is Institute of Molecular Biotechnology of the Austrian Academy of Sciences (IMBA), Vienna Biocenter, Vienna, Austria
*Sebastian Wild, Aleksandra Lezaja, and Federico Teloni contributed equally to this work

hydrolysis, and helicase activities (Capo-Chichi et al, 2013). Cells derived from these patients also display cohesion defects, but they seem to be less pronounced than in patients with mutations that prevent expression of a stable full-length protein (Capo-Chichi et al, 2013; Alkhunaizi et al, 2018), further suggesting that DDX11 influences cohesion establishment in a helicase-dependent and helicase-independent manner.

Here, we show that coordination of an FeS cluster is required for all of DDX11's biochemical activities and that residue R263 impacts on FeS cluster binding, most likely because of its positive charge. We further show that DDX11 interacts with DNA polymerase delta (Pol δ) and—like its yeast homologue (Samora et al, 2016)—with WDHD1/hCTF4. In vitro, DDX11 can remove DNA obstacles ahead of Pol δ in an ATPase- and FeS cluster domain-dependent manner, and hence generate ssDNA. In agreement with these findings, we show depletion of DDX11 to cause reduced levels of ssDNA and chromatin-bound replication protein A (RPA) and to impair CHK1 phosphorylation at serine-345 (CHK1-pS345). Taken together, we propose that DDX11 promotes the generation of ssDNA by dismantling secondary structures during DNA replication and thereby contributes to CHK1 activation.

# Results

## DDX11 coordinates an FeS cluster that is required for all of its biochemical activities

DDX11 has been proposed to bind an FeS cluster because of its homology with XPD (Rudolf et al, 2006), but experimental evidence is missing so far. The putative FeS cluster-binding motif resides in its helicase domain and contains four strictly conserved cysteines (Fig S1A and B). Using a radioactive iron incorporation assay, we observed incorporation of iron-55 into wild-type DDX11, suggesting that DDX11 can indeed bind an FeS cluster (Figs 1A and S1C and D). In contrast, when the first or fourth cysteine of the FeS cluster-binding motif was replaced by serine (C267S and C350S DDX11, respectively), iron incorporation was decreased to background levels (Figs 1A and S1C and D), underlining the importance of these residues for FeS cluster coordination.

To investigate the importance of the FeS cluster for the biochemical activities of DDX11, we next purified C-terminally Flag-tagged DDX11 variants from Sf9 insect cells (Fig 1B). Using electrophoretic mobility shift assays (EMSAs), we observed that both cysteine variants were unable to bind to a variety of oligo-based DNA substrates (Figs 1C and S1E). Moreover, they displayed similar defects in DNA-stimulated ATP hydrolysis (Fig 1D) and helicase activity (Fig 1E) as the ATPase-dead variant K50R DDX11 (Hirota & Lahti, 2000).

## The WABS-associated variant R263Q is deficient in FeS cluster binding

The WABS-associated variant R263Q was shown to display defects in DNA binding, ATP hydrolysis and helicase activity (Capo-Chichi et al, 2013), but it is unclear whether these defects stem from impaired FeS cluster binding. To address this question, we used the same iron incorporation assay as above and observed a greatly reduced incorporation of iron-55

by the R263Q variant (Figs 2A and S2A). We then wondered whether the positive charge of R263 was required for stable FeS cluster incorporation and tested FeS cluster binding by the two rationally designed variants R263K and R263E, in which arginine was replaced either by the positively charged lysine or the negatively charged glutamic acid. Whereas in the R263K variant FeS cluster binding was partially restored, R263E displayed very low iron incorporation (Figs 2A and S2A) that was comparable with the FeS cluster binding–deficient cysteine variants. Moreover, in the R263K variant DNA binding, ATP hydrolysis and DNA unwinding were restored or partially restored, whereas R263E—like the patient variant R263Q—had no detectable biochemical activities (Fig 2B–E), suggesting that it is the positive charge at position 263 that is required for stable FeS cluster incorporation.

Taken together, our data so far establish that FeS cluster binding is required for all of DDX11's biochemical activities, and that deficient FeS cluster binding most likely causes the biochemical defects displayed by the WABS-associated variant R263Q.

## DDX11 interacts with factors involved in DNA replication independently of its FeS cluster

DDX11 has been shown to co-localise with sites of DNA synthesis (Cortone et al, 2018) and to interact with proteins found at the replication fork, such as proliferating cell nuclear antigen (PCNA), FEN1, and Timeless (Farina et al, 2008; Leman et al, 2010; Cortone et al, 2018). Moreover, the S. cerevisiae homologue of DDX11, Chl1, was previously shown to be recruited to the replication fork by Ctf4 through a Ctf4-interacting peptide (CIP) box (Samora et al, 2016). Because a previous study made mention of only a weak interaction between human DDX11 and CTF4/WDHD1 (Farina et al, 2008), and no obvious CIP box motif is found in human DDX11, it has, however, remained unclear whether the interaction between DDX11 and WDHD1 is conserved in humans.

To further define DDX11's interaction network, we performed a pull-down experiment with YFP-tagged DDX11 from HeLa Flp-In T-REx cells and analysed associated proteins by mass spectrometry. In agreement with previous findings (Farina et al, 2008; Leman et al, 2010; Cortone et al, 2018), Gene Ontology term enrichment analysis revealed that DNA replication and the mitotic cell cycle were amongst the pathways that were most enriched (Fig 3A, Tables S1, and S2).

Interestingly, WDHD1 was amongst YFP-DDX11's putative interaction partners (Tables S1 and S2) and could be pulled down with Flag-tagged DDX11 from 293T cells (Fig 3B). The interaction could be further corroborated by reciprocal immunoprecipitation (IP) experiments (Fig S3A). Despite the lack of an obvious CIP box in human DDX11, the interaction between DDX11 and WDHD1, hence, seems to be conserved.

Moreover, we noticed that—with DNA polymerase alpha and delta—two additional proteins involved in lagging-strand DNA replication were found in DDX11's interactome (Tables S1 and S2). Endogenous POLD1, the catalytic subunit of Pol δ, was also pulled down with Flag-tagged DDX11 from 293T cells (Fig 3B). To further confirm this interaction, we then performed co-IP experiments with over-expressed DDX11 and POLD1 in 293T cells that revealed a reciprocal interaction between the two proteins (Fig 3C). Importantly, these interactions were not mediated by DNA because extracts were routinely treated with Benzonase nuclease.

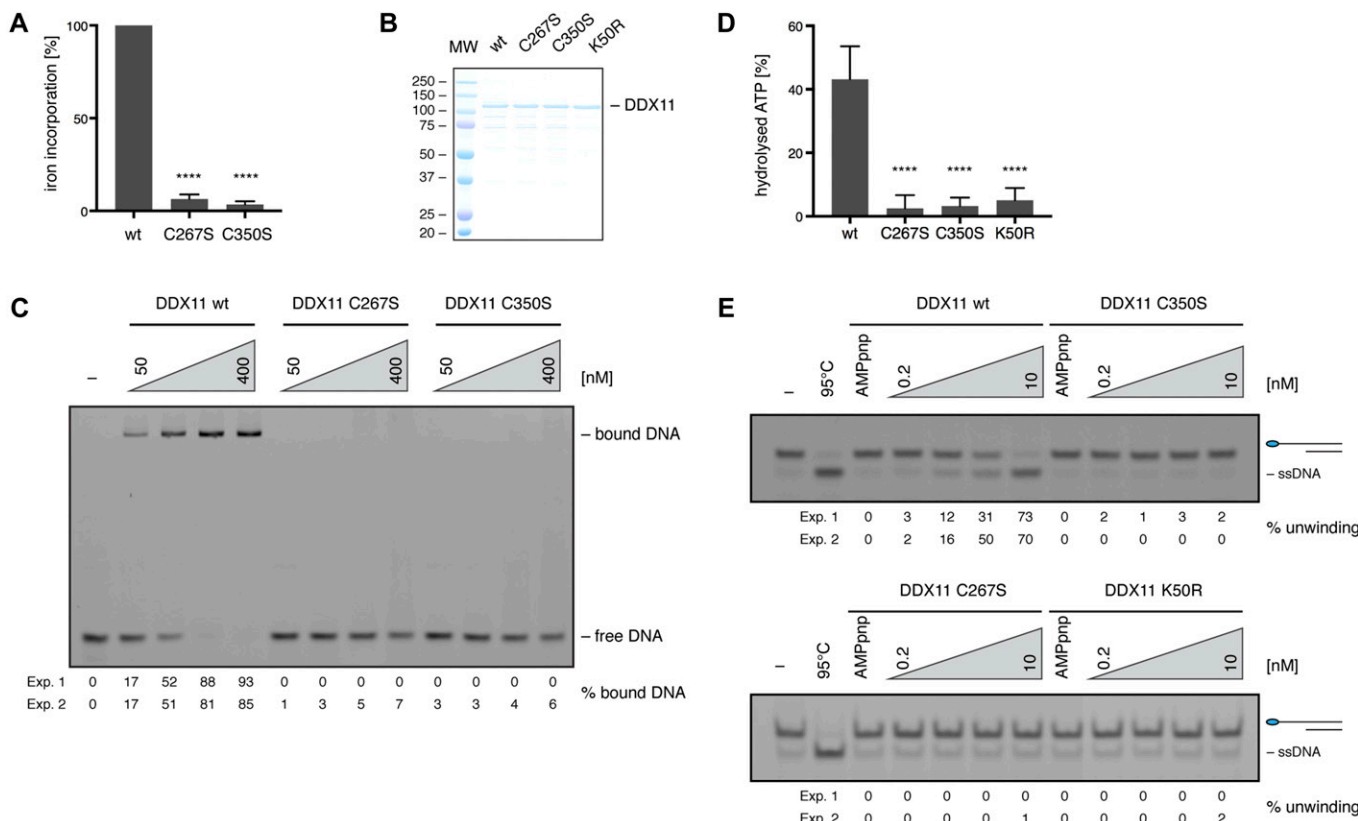

**Figure 1. FeS cluster binding is indispensable for DDX11's biochemical activities.**
**(A)** Radioactive iron-55 incorporation in wild-type (wt) DDX11 and cysteine variants, as measured by liquid scintillation counting. Levels are expressed as % iron incorporation, with wild-type levels set to 100%. Error bars depict standard deviations from three independent experiments. Statistical analysis: ordinary one-way ANOVA (****$P < 0.0001$). **(B)** InstantBlue-stained SDS gel of purified DDX11 variants. **(C)** Electrophoretic mobility shift assays with 10 nM of FAM-labelled 5'-overhang substrate and increasing amounts of DDX11 variants. Numbers indicate the percentage of bound DNA from two independent experiments (Exp. 1 and Exp. 2). **(D)** ATPase activity of DDX11 variants in the presence of single-stranded DNA, as measured by the release of inorganic phosphate from radio-labelled $\gamma$-$^{32}$P-ATP in TLC. Activity is depicted as % of hydrolysed ATP, with background activity in the absence of DNA subtracted. Error bars depict standard deviations from three independent experiments. Statistical analysis: ordinary one-way ANOVA (****$P < 0.0001$). **(E)** Helicase assays with 10 nM of FAM-labelled 5'-overhang substrate and increasing amounts of DDX11 variants. Numbers indicate the percentage of unwound DNA from two independent experiments (Exp. 1 and Exp. 2). See also Fig S1. MW, molecular weight. Source data are available for this figure.

Because in other proteins loss of the FeS cluster has been reported to abolish interaction with partner proteins (Netz et al, 2012; Vashisht et al, 2015), we also tested whether the WABS variant R263Q and the FeS cluster binding–deficient variants C267S and C350S were impaired in their interaction with POLD1 and WDHD1. In co-IP experiments, all variants were, however, able to interact with both POLD1 (Fig 3D) and WDHD1 (Fig S3B), suggesting that FeS cluster binding is not required for the interaction of DDX11 with its partner proteins. This notion was further confirmed by co-IP experiments with WDHD1 and DDX11 fragments, which showed an interaction of WDHD1 with the C-terminal part of DDX11 that is at a distance of the FeS domain (Fig S3C and D).

### DDX11 can remove DNA obstacles ahead of DNA polymerase delta

Given DDX11's interactions with lagging-strand proteins and its helicase activity, a role for DDX11 in the resolution of secondary structures during DNA replication has been discussed (Bharti et al, 2014; Pisani et al, 2018). To directly address whether DDX11 may be able to functionally interact with Pol δ, for example, by removing DNA obstacles from its DNA template, we performed time-resolved

primer extension assays with purified Pol δ and DDX11 variants (Fig 4). To mimic a replication block (which DNA secondary structures would impose), we annealed a 15-nucleotide (nt)-long 3'-bio-tinylated DNA oligo ahead of the primer (Fig 4A). In contrast to a substrate without DNA block, in which the primer could be readily elongated by Pol δ (Fig 4B, left), such a DNA block constituted an effective roadblock for DNA synthesis (Fig 4B, second from left). In the presence of wild-type DDX11, however, Pol δ was able to extend the primer to a similar extent to what was observed without a DNA block (Fig 4B, third from left). Importantly, primer removal by DDX11 was dependent on a functional ATPase domain and an intact FeS cluster because addition of neither the WABS variant R263Q nor the FeS cluster–deficient or ATPase-dead variants could confer primer extension by Pol δ (Fig 4B and C).

### DDX11 promotes the generation of ssDNA

Unwinding of DNA secondary structures in the context of lagging-strand DNA replication may expose ssDNA. We, thus, reasoned that loss of DDX11 should be accompanied by a reduction in exposed

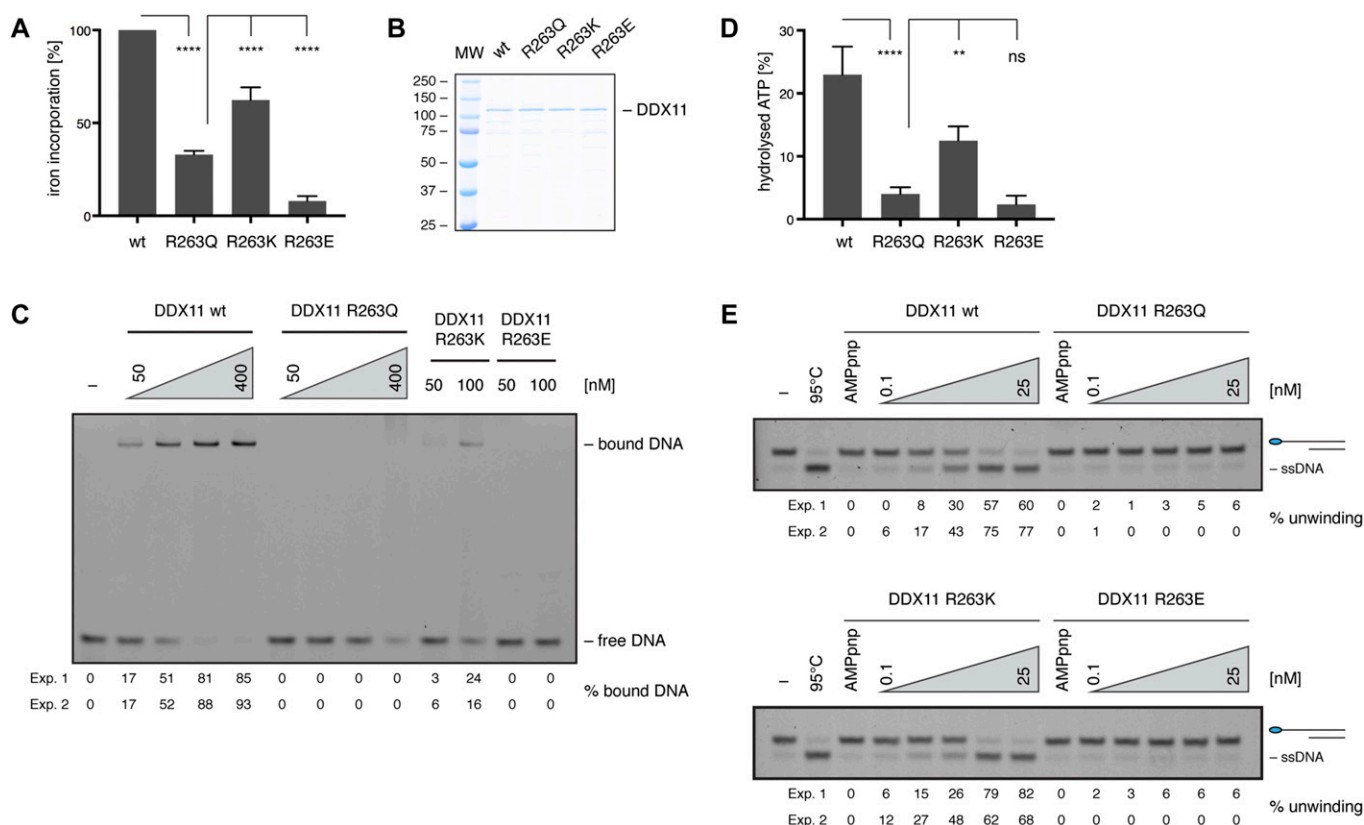

**Figure 2. The positive charge at position 263 is important for FeS cluster binding and DDX11's biochemical activities.**
**(A)** Radioactive iron-55 incorporation in wild-type (wt) DDX11 and arginine variants, as measured by liquid scintillation counting. Levels are expressed as % iron incorporation, with wild-type levels set to 100%. Error bars depict standard deviations from three independent experiments. Statistical analysis: ordinary one-way ANOVA (****$P$ < 0.0001). **(B)** InstantBlue-stained SDS gel of purified DDX11 variants. **(C)** Electrophoretic mobility shift assays with 10 nM of FAM-labelled 5'-overhang substrate and increasing amounts of DDX11 variants. Numbers indicate the percentage of bound DNA from two independent experiments (Exp. 1 and Exp. 2). **(D)** ATPase activity of DDX11 variants in the presence of single-stranded DNA, as measured by release of inorganic phosphate from radio-labelled γ-$^{32}$P-ATP in TLC. Activity is depicted as % of hydrolysed ATP, with background activity in the absence of DNA subtracted. Error bars depict standard deviations from three independent experiments. Statistical analysis: ordinary one-way ANOVA (****$P$ < 0.0001; **$P$ = 0.0045; ns, nonsignificant). **(E)** Helicase assays with 10 nM of FAM-labelled 5'-overhang substrate and increasing amounts of DDX11 variants. Numbers indicate the percentage of unwound DNA from two independent experiments (Exp. 1 and Exp. 2). See also Fig S2. MW, molecular weight.

ssDNA if DDX11 were indeed to play a role in the removal of replication-associated DNA secondary structures. To exacerbate the formation of secondary structures during DNA replication, we evoked conditions of enhanced replicative helicase/polymerase uncoupling by treating RPE-1 cells for 90 min or 2 h with hydroxyurea (HU), which inhibits ribonucleotide reductase and, thereby, depletes the nucleotide pool, and the ataxia telangiectasia and Rad3-related protein (ATR) inhibitor VE-821 (ATRi). In these conditions, using quantitative image-based cytometry (QIBC) (Toledo et al, 2013; Teloni et al, 2019), we observed a reduction in native ssDNA upon depletion of DDX11, specifically in S-phase cells (Fig 5A and C).

This reduction was accompanied by reduced levels of the ssDNA-binding protein RPA on chromatin (Fig 5B and C). Accordingly, in cell fractionation experiments, RPA levels on chromatin were lower upon depletion of DDX11, both in untreated conditions and in response to a treatment with HU and ATRi (Fig 5D).

In conclusion, our data suggest a role for DDX11 in the generation of ssDNA, which we surmise is due to its ability to remove DNA secondary structures during DNA replication.

## DDX11 is required for CHK1 phosphorylation at serine-345

Under conditions of replication stress, RPA-coated ssDNA is an important intermediate for ATR-dependent checkpoint activation (Saldivar et al, 2017). Because we observed a reduction of ssDNA and chromatin-bound RPA upon *DDX11* knock-down, we wondered whether phosphorylation of the ATR effector kinase CHK1 at serine-345 was affected in the absence of DDX11. To address this question, we treated RPE-1 cells for 30 min to 4 h with HU. In contrast to control cells where such a treatment led to a robust phosphorylation of CHK1, DDX11-depleted cells displayed only a mild induction of CHK1-pS345 (Figs 6A and S4A). Moreover, also in response to aphidicolin (Aph), an inhibitor of replicative DNA polymerases, and camptothecin (CPT), a topoisomerase 1 inhibitor, CHK1 was less phosphorylated at serine-345 (Fig 6B and C).

Interestingly, also the levels of total CHK1 were reduced by depletion of DDX11, even in untreated (0 h) conditions (Fig 6A–C). This may be explained by recent work establishing the need for a basal CHK1 activity during unchallenged conditions to prevent CHK1

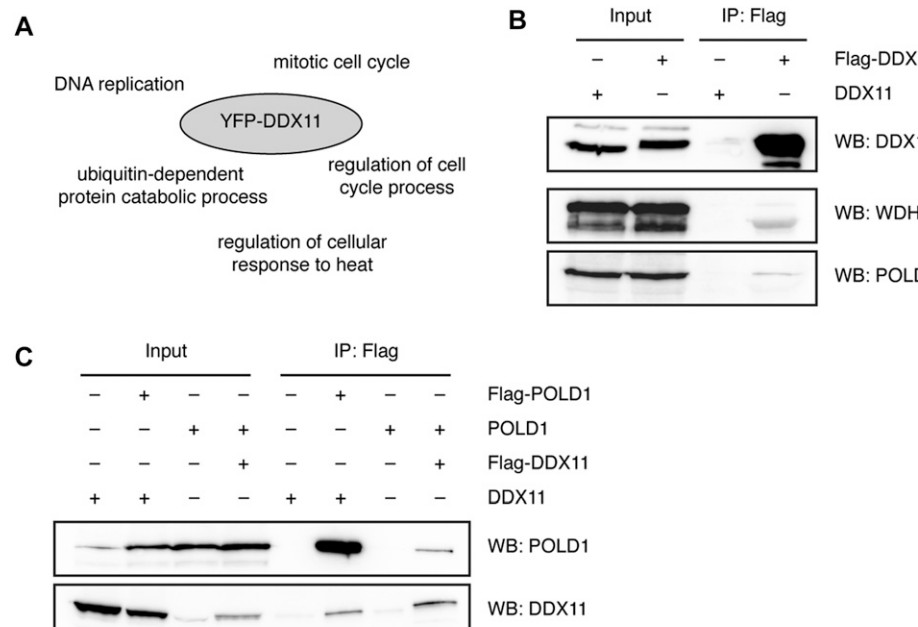

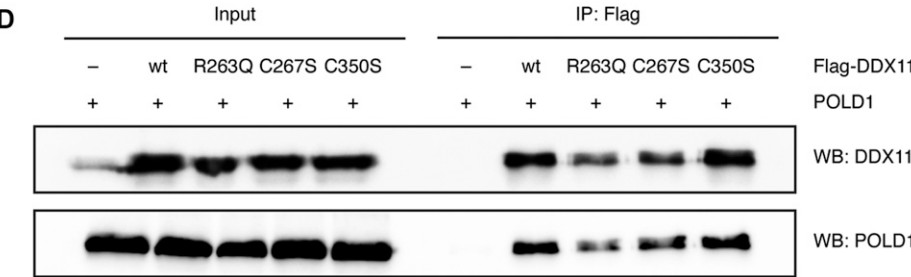

**Figure 3. DDX11 interacts with Pol δ independently of its FeS cluster.**
**(A)** Gene Ontology term enrichment analysis of interaction partners obtained upon pull-down of YFP-DDX11 from HeLa Flp-In T-REx cells. **(B)** Flag pull-down of over-expressed Flag-tagged DDX11 from 293T cells and co-immunoprecipitated endogenous proteins. **(C)** Reciprocal co-immunoprecipitations of Flag-tagged POLD1 and untagged DDX11, and Flag-tagged DDX11 and untagged POLD1, respectively, extracted from 293T cells. **(D)** Co-immunoprecipitations of Flag-tagged DDX11 variants and untagged POLD1 from 293T cells. See also Fig S3, Tables S1, and S2. WB, Western blot.

degradation (Michelena et al, 2019). We suspect that DDX11 may be required for CHK1's steady-state activity and that loss of DDX11 may in turn affect CHK1 stability.

Taken together, our data establish that the reduction of ssDNA and chromatin-bound RPA that we observe upon depletion of DDX11 in response to DNA replication stress is accompanied by a reduced phosphorylation of CHK1 at serine-345.

## Discussion

Here, we show that DDX11 coordinates an FeS cluster that is required for DNA binding and, presumably as a consequence thereof, for DNA-dependent ATPase activity and DNA unwinding. Loss of the FeS cluster in DDX11, hence, seems to cause a more severe defect than in the related helicase XPD/Rad3, in which FeS cluster loss was found to only impair DNA unwinding, whereas DNA binding and ATP hydrolysis were largely unaffected (Rudolf et al, 2006; Pugh et al, 2008). Because there is no crystal structure of DDX11 available, we can only speculate about this discrepancy. From structural studies with *Sulfolobus acidocaldarius* XPD (SaXPD) and two further Archaea homologues, it would appear that the FeS domain in XPD forms a narrow tunnel with

the neighbouring arch domain and, hence, contributes to ssDNA binding, but that the DNA binding channel is long enough to compensate for a loss of grip in the absence of an FeS cluster (Fan et al, 2008; Liu et al, 2008; Wolski et al, 2008). In contrast, the DNA binding channel may be shorter in DDX11, thus rendering a properly folded FeS domain essential for DNA binding.

Despite these differences, lysine-84 in SaXPD was shown to stabilise FeS cluster binding (Fan et al, 2008), similar to what we see for the homologous residue arginine-263 in DDX11, suggesting that FeS cluster coordination symmetry is similar between the two proteins. Interestingly, this stabilisation was shown to be mediated by lysine-84's ability to form a charged hydrogen bond to one of the cysteine ligands of the FeS cluster (Fan et al, 2008), which is in agreement with our finding that re-introduction of a positive charge at position 263 (R263K variant) can partially revert the FeS cluster-binding defects and impaired biochemical activities observed with the WABS variant R263Q. The importance of this conserved arginine/lysine residue in FeS cluster coordination is further underlined by the fact that the homologous residue in human XPD (arginine-112) is required for helicase activity (Dubaele et al, 2003; Rudolf et al, 2006) and affected in the XPD-linked disorder trichothiodystrophy (Botta et al, 2002).

Although loss of the FeS cluster in DDX11 most probably causes a local unfolding of the FeS domain that affects DNA binding, it does

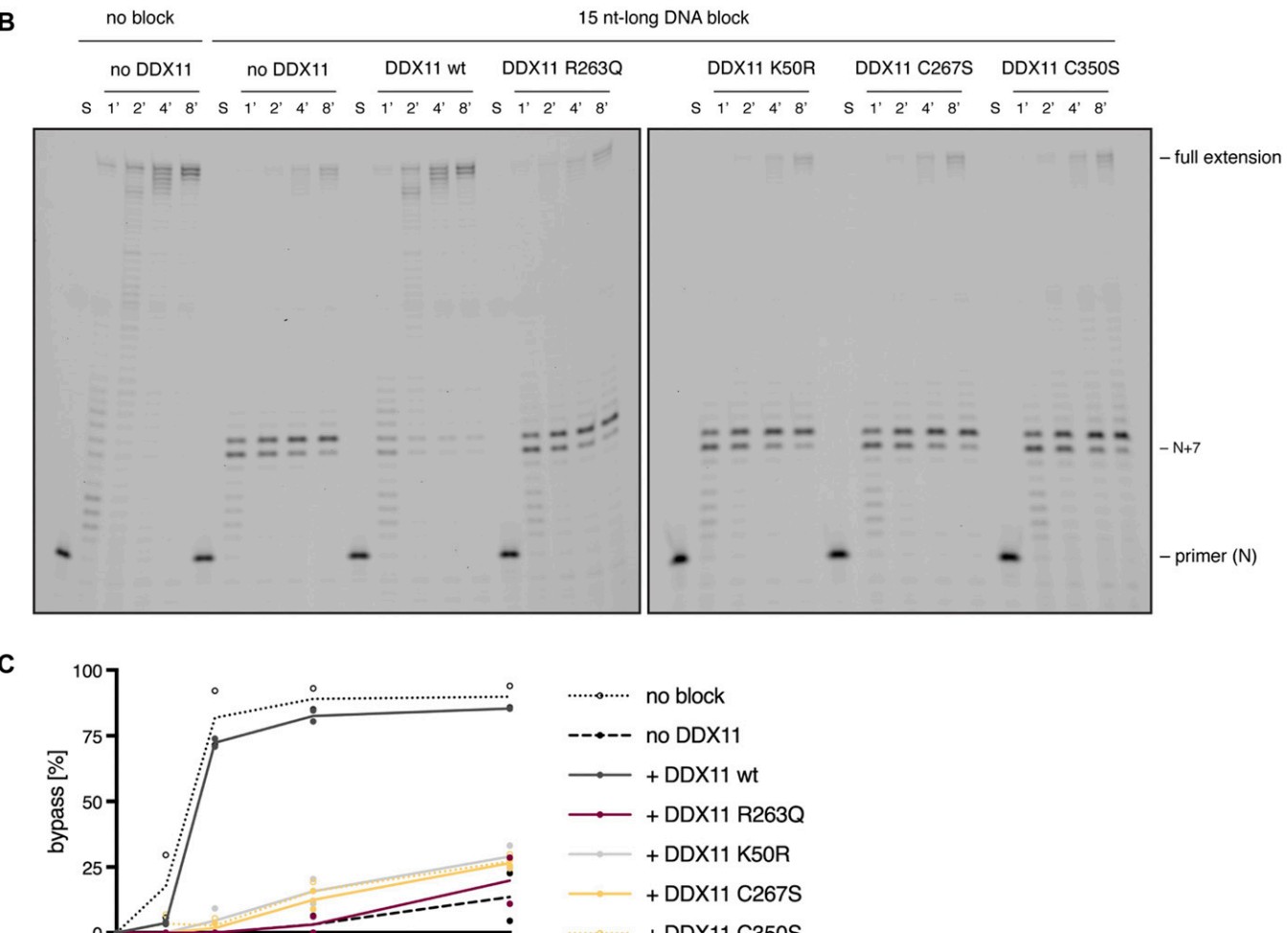

**Figure 4. DDX11 can remove obstacles from the DNA template ahead of Pol δ.**
**(A)** Schematic of primer extension assay. Blue ellipse depicts 5′-FAM label on the primer that gets extended by Pol δ. Light grey circle depicts 3′-biotin on the DNA block that was added to prevent primer extension. Numbers indicate lengths of primers and gaps in nucleotides. **(B)** Time-resolved primer extension assay with substrate as depicted in A (15 nt-long DNA block) or without a DNA obstacle (no block). S denotes substrate only. In all other lanes, 2.5 nM of Pol δ was added in the absence or presence (25 nM) of DDX11 variants. N+7 denotes extension of primer by seven nucleotides. **(C)** Quantification of primer extension beyond the DNA block in the absence or presence of 25 nM DDX11 variants. Data points indicate values from two independent experiments. Lines connect the theoretical mean values.

not seem to lead to a complete unfolding of the protein because all variants with alterations in the FeS cluster-binding pocket are still able to interact with Pol δ and WDHD1. Interestingly, Pol δ and WDHD1 join a group of DDX11 interaction partners, namely, PCNA, Timeless, and FEN1 (Farina et al, 2008; Leman et al, 2010; Cortone et al, 2018), which all play a role in lagging-strand DNA replication, rendering a function of DDX11 in this context highly likely. In agreement with this idea, we find DDX11 to be able to functionally interact with Pol δ in vitro. Indeed, DDX11 is able to remove a DNA obstacle from the DNA template ahead of Pol δ and to allow Pol δ to extend a primer to the same extent as if no roadblock were present. Importantly, this function is dependent on DDX11's ATPase and FeS

cluster-binding domains, suggesting that one of the helicase-dependent functions of DDX11 could be the dismantling of secondary DNA structures, as proposed previously (Bharti et al, 2014; Pisani et al, 2018).

To exaggerate the formation of replication-dependent secondary structures in vivo and to address whether DDX11 could play a role in their resolution, we treated cells with HU and ATRi. In the absence of DDX11, such a treatment led to reduced levels of native ssDNA and chromatin-bound RPA. Although these findings are in agreement with a role for DDX11 in the unwinding of DNA secondary structures, future studies will have to demonstrate whether DDX11's helicase activity or an intact FeS domain are indeed required for

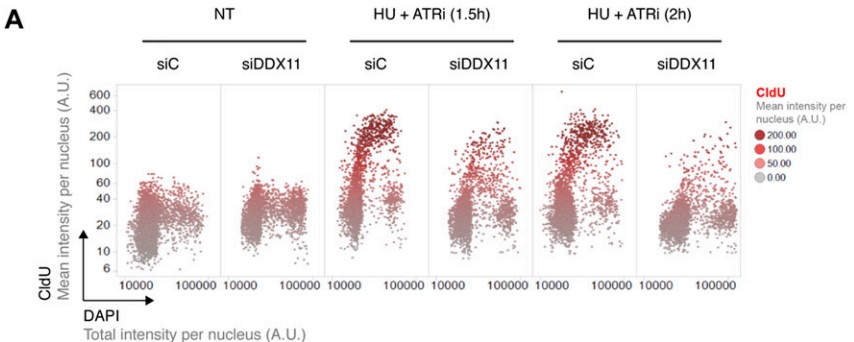

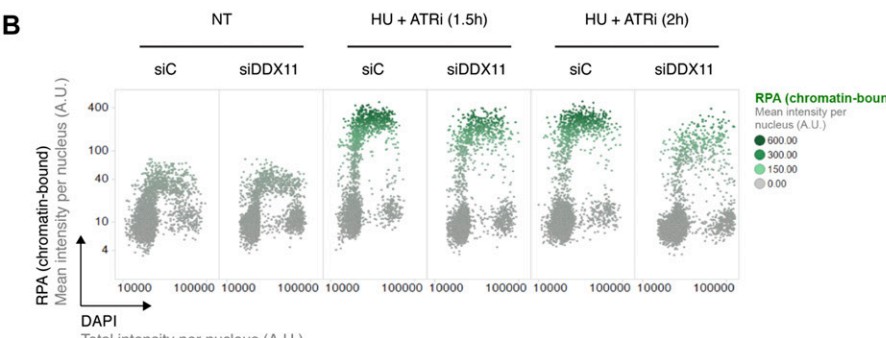

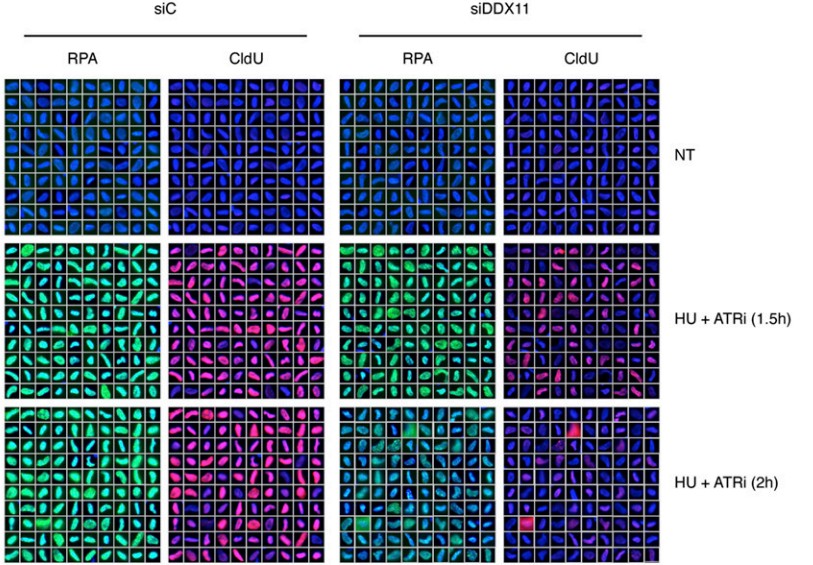

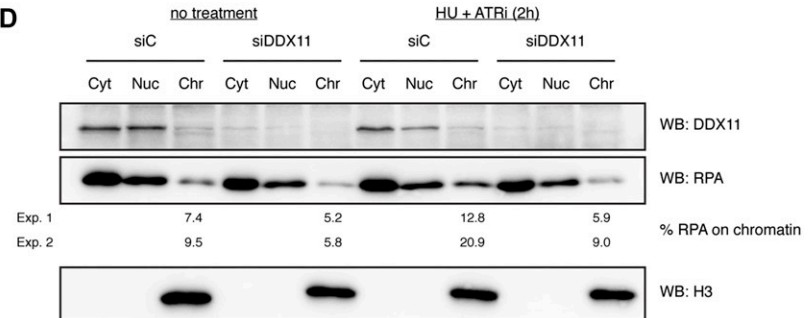

**Figure 5. DDX11 promotes the formation of single-stranded DNA.**
**(A, B)** QIBC experiment showing the mean intensity per nucleus of native CldU indicative of single-stranded DNA (A) or chromatin-bound RPA (B) plotted against the total DAPI intensity per nucleus. The RPE-1 cells had been labelled for 24 h with the nucleotide analogue CldU followed by 1 h 30 min or 2 h of treatment with 2 mM HU and 2 µM ATRi. Control cells were left non-treated. The cells were pre-extracted before fixation. Levels of CldU and RPA are colour-coded, as indicated in the legends. **(C)** Software-based random selection of S-phase cells, as defined by an intermediate DNA content (>2N and <4N) and RPA positivity. Overlays of DAPI and the chromatin-bound RPA and native CldU signals are shown. Scale bar, 20 µm. **(D)** Cell fractionation into cytoplasmic (Cyt), nucleoplasmic (Nuc), and chromatin (Chr) extracts in untreated cells (left) or cells treated with 2 mM HU and 2 µM ATRi for 2 h (right). Histone H3 was used as a chromatin marker. Numbers indicate the percentage of RPA on chromatin from two independent experiments (Exp. 1 and Exp. 2). A.U., arbitrary units; siC, siControl; WB, Western blot.

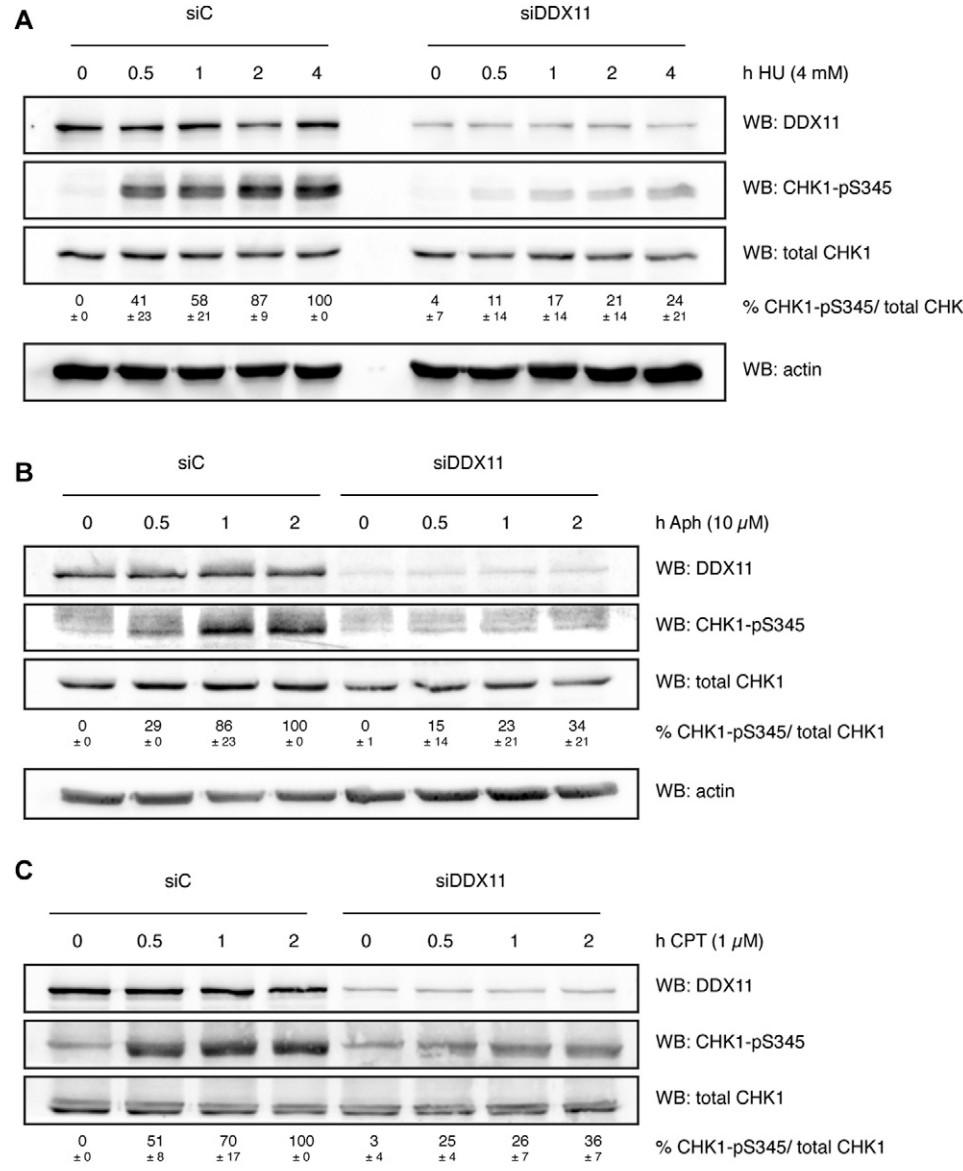

**Figure 6. DDX11 is required for CHK1 activation.**
**(A, B, C)** Western blots showing time course of CHK1-pS345 activation in control RPE-1 cells (siC) and cells depleted of DDX11 (siDDX11). **(A, B, C)** Cells were treated or not with 4 mM HU (A), 10 μM aphidicolin (Aph) (B), and 1 μM camptothecin (CPT) (C). Numbers indicate the mean values and standard deviations of the percentage of CHK1-pS345 per total CHK1 from three independent experiments. See also Fig S4. WB, Western blot.

this function. Support for this idea comes from a very recent study that establishes a helicase-dependent role for yeast DDX11, Chl1, in cohesin loading and RPA accumulation at HU-arrested replication forks (Delamarre et al, 2019).

RPA-coated ssDNA is an important activator of ATR-dependent checkpoint signalling, which ultimately leads to the stabilisation of active replication forks, the suppression of new origin firing, and delayed cell cycle progression (Saldivar et al, 2017). In line with our observation that DDX11 promotes the generation of ssDNA and the loading of RPA on chromatin, phosphorylation of the ATR effector kinase CHK1 at serine-345 was greatly impaired upon DDX11 depletion in response to a variety of replication stress-inducing agents, suggesting a role for DDX11 in CHK1 activation. These findings are in agreement with a previous study, in which knock-down of *DDX11*—in

addition to a slight reduction in CHK1-pS345—caused impaired fork restart and increased origin firing upon HU treatment (Calì et al, 2016). Because functional CHK1 signalling is required to prevent entry into mitosis under conditions of replication stress (Saldivar et al, 2017), a function of DDX11 in CHK1 activation could also explain—at least in part—why DDX11 depletion is accompanied by defects in chromosome segregation (Parish et al, 2006). Surprisingly, however, the function of DDX11 in CHK1 activation does not seem to be conserved in budding yeast and DT-40 cells where phosphorylation of CHK1 was unaffected by the absence of DDX11 (Laha et al, 2006; Abe et al, 2018).

In human cells, DDX11 seems to have roles in cohesion establishment that are both helicase-dependent and helicase-independent because an ATPase-dead version of DDX11 can partially restore cohesion

establishment in DDX11-depleted HeLa cells (Cortone et al, 2018). Taking into account previous data and our own, we propose a dual function for DDX11 in cohesion establishment: first, a direct role in the loading or the retention of cohesin on chromatin (Borges et al, 2013; Cortone et al, 2018) that remains to be further defined, and second, an indirect, most likely helicase-dependent, role through its ability to activate CHK1. Such a dual function would be in agreement with the observation that WABS patients who have wild-type levels of the biochemically inactive R263Q variant have a less severe cohesion phenotype than patients who have barely detectable DDX11 (Capo-Chichi et al, 2013; Alkhunaizi et al, 2018). Interestingly, however, these patients display similar congenital abnormalities as other WABS patients (Capo-Chichi et al, 2013), which may suggest that the helicase-dependent function of DDX11 is particularly important during embryonic development.

# Materials and Methods

### Plasmids and primers

*WDHD1* and *POLD1* cDNAs were purchased from GE Healthcare Dharmacon (IMAGE ID: 6514641 and 3634655, respectively) and used as a PCR template. *DDX11* cDNA was amplified from pcDNA3-6xHis-DDX11-3xFlag (Wu et al, 2012). All cDNAs were inserted into the GATEWAY entry vector pDONR221 (Thermo Fisher Scientific) according to the manufacturer's instructions. To insert point mutations in entry clones, site-directed mutagenesis was performed after the QuikChange Site-Directed Mutagenesis manual (Stratagene). All entry clones were checked by sequencing and used to shuttle the gene of interest into destination vectors.

### Sf9 insect cells and baculoviruses

*Sf9* insect cells were grown in HyClone SFX-insect medium (GE Healthcare), without supplements and incubated at 27°C either as a monolayer culture or in suspension. Bacmids and baculoviruses were generated using the Bac-to-Bac Baculovirus Expression System (Invitrogen). To express a protein of interest, $2 \times 10^6$ cells/ml were infected at an estimated MOI of 1, and incubated for 48 h at 27°C on a horizontal shaker at 200 rpm. Then, the cells were harvested by centrifugation at 475$g$ for 10 min.

### Protein purification

For lysis, *Sf9* pellets over-expressing *DDX11-C-3xFLAG* were resuspended in 3–4 pellet volumes of ice-cold C1 buffer (25 mM Tris [pH 7.6], 150 mM NaCl, 10% glycerol, 1.5 mM MgCl$_2$, 1 mM EDTA, and 0.5 mM $\beta$-mercaptoethanol) supplemented with 0.2% NP-40 and cOmplete EDTA-free protease inhibitor cocktail (Roche). Lysates were incubated on ice for 30 min and vortexed occasionally. To clarify the lysate, cell membranes and debris were sedimented by centrifugation at 100,000$g$ and 4°C for 30 min in an ultracentrifuge WX Ultra Series (Sorvall) equipped with a T-865 rotor (Sorvall), followed by filtration of the supernatant with a syringe filter of 0.45-$\mu$m poresize. The filtered lysate was incubated with equilibrated ANTI-FLAG M2 Affinity Gel (Sigma-Aldrich) (200 $\mu$l per 100 ml starting culture)

overnight at 4°C with rotation. The next day, the beads were spun down by centrifugation at 1,000$g$ and 4°C and the supernatant was removed. The beads were washed one time quickly and three times for 15 min with C1 buffer supplemented with 0.1% NP-40, followed by one quick and one 15-min wash with C1 buffer without NP-40. Afterwards, DDX11 was eluted from the beads by incubation with 2–5 volumes of C1 buffer supplemented with 200 ng/$\mu$l 3xFLAG peptide (Sigma-Aldrich) for 2 h at 4°C with rotation. Quantity and purity of the preparations were assessed by SDS–PAGE and InstantBlue stain.

Pol δ in complex with PCNA was produced in *Sf9* insect cells and purified via the Flag-tag on POLD1, as described previously (Jozwiakowski et al, 2019).

### 55-iron incorporation assay

A 20-ml culture of *Sf9* cells at $2 \times 10^6$ cells/ml was infected with the respective baculoviruses at an MOI of 1 and supplemented with 200 $\mu$l of 0.1 M sodium ascorbate and 20 $\mu$l of $^{55}$FeCl$_3$ (1 mCi/ml). After 48 h of incubation, the cells were harvested by centrifugation for 10 min at 475$g$ and 4°C. Supernatant was removed and the cell pellets were washed once with 5 ml of citrate buffer (50 mM citrate and 1 mM EDTA in PBS, pH 7.0) and once with 10 ml PBS. Next, the cell pellet was resuspended in 1 ml of cold C1 buffer, supplemented with 0.1% NP-40 and protease inhibitor cocktail (Roche) and transferred to a 2-ml Eppendorf tube. The lysate was incubated on ice for 30 min. After centrifugation for 30 min at 17,000$g$ and 4°C, the clarified lysate was transferred to a second tube containing 20 $\mu$l of pre-equilibrated ANTI-FLAG M2 Affinity Gel (Sigma-Aldrich). For protein capture, lysate and beads were incubated for 2 h on a rotator at 4°C. Then, the beads were washed 6× with 1 ml C1 lysis buffer supplemented with 0.1% NP-40. Meanwhile, liquid scintillation tubes were prepared by adding 1 ml of Ultima Gold liquid scintillation liquid (PerkinElmer) to the counting tubes. The washed beads were resuspended in 50 $\mu$l of H$_2$O, and 50% of the beads suspension was transferred into the scintillation liquid and vortexed extensively. The cpm were measured with a Tri-Carb scintillation counter (Packard) using standard $^3$H settings. The remaining beads were taken up in 2× sample buffer and run on an SDS–PAGE. Protein levels were quantified using the InstantBlue-stained gel and ImageJ software (Schneider et al, 2012) and used to normalise counts.

### DNA substrates

Oligonucleotides were ordered at Microsynth, either unlabelled or with a fluorescein amidite (FAM) label at the 5′-end. DNA substrates for EMSAs and DNA helicase assays were generated by annealing 200 nM of FAM-labelled oligonucleotide and 300 nM of unlabelled oligonucleotide(s) in 10 mM Tris–HCl (pH 8.0), 50 mM NaCl, and 10 mM MgCl$_2$ in a PCR cycler that was programmed to heat the sample to 95°C for 5 min, followed by a step-wise temperature reduction to 20°C (−5K every 3 min). DNA substrates for primer extension assays were prepared the same, except that 200 nm of FAM-labelled primers were annealed with 200 nM of template oligonucleotide and 400 nM of 3′-biotinylated block oligonucleotide in 10 mM Tris–HCl (pH 8.0), 50 mM KCl, and 0.5 mM EDTA.

**Primers used for site-directed mutagenesis.**

| Site-directed mutagenesis primer | Sequence (5′–3′) |
|---|---|
| K50R (forward) | CCAACTGGCACTGGG**AGG**TCCTTAAGTC |
| K50R (reverse) | GACTTAAGGA**CCT**CCCAGTGCCAGTTGG |
| R263Q (forward) | GGTCTCCCTTGGCTCC**CAG**CAGAACCTTTG |
| R263Q (reverse) | CAAAGGTTCTG**CTG**GGAGCCAAGGGAGACC |
| R263K (forward) | GGTCTCCCTTGGCTCC**AAG**CAGAACCTTTGTG |
| R263K (reverse) | CACAAAGGTTCTG**CTT**GGAGCCAAGGGAGACC |
| R263E (forward) | GGTCTCCCTTGGCTCC**GAG**CAGAACCTTTGTG |
| R263E (reverse) | CACAAAGGTTCTG**CTC**GGAGCCAAGGGAGACC |
| C267S (forward) | CCCGGCAGAACCTT**AGT**GTAAATGAAGACGTG |
| C267S (reverse) | CACGTCTTCATTTAC**ACT**AAGGTTCTGCCGGG |
| C350S (forward) | GAGGCCCGGGCC**AGT**CCCTATTACGGG |
| C350S (reverse) | CCCGTAATAGGG**ACT**GGCCCGGGCCTC |

## EMSA

10-μl reactions were set up containing 10 nM of FAM-labelled substrate, 3 μl of purified protein or protein dilution, 0.1 mg/ml BSA, 5 mM EDTA, and 1 mM DTT. The reactions were incubated on ice for 30 min before adding 10 μl 2× EMSA loading dye (7% Ficoll, 20 mM Tris–HCl [pH 8.0], and 20 mM EDTA). Protein–DNA complexes were separated on a 10% non-denaturing polyacrylamide gel (1× ̂γTris-borate-EDTA [TBE] with 19:1 acrylamide to bis-acrylamide ratio) in cold 0.5× TBE buffer at 80 V for 1 h 30 min. For signal detection, the gel was scanned with a Typhoon FLA9500 laser scanner (GE Healthcare) set to the fluorescence-imaging mode.

## DNA helicase assay

Helicase assay reactions contained 0.1 mg/ml BSA, 1 mM DTT, 25 mM potassium acetate, 10 nM FAM-labelled substrate, and 3 μl protein or protein dilution. A 9-μl mix of substrate and protein was pre-incubated for 15 min on ice, before adding 1 μl of 10× ATP mix to obtain final concentrations of 1 mM ATP, 0.5 mM $MgCl_2$, and 50 nM competitor DNA. Reactions were incubated for 15 min at 37°C and stopped by the addition of 10 μl 2× helicase loading dye (7% Ficoll, 20 mM Tris–HCl pH 8.0, 20 mM EDTA, 2 mg/ml Proteinase K [Fermentas]) and a further 10-min incubation at 37°C. The samples were analysed as described for EMSAs.

## ATPase assay

ATPase activity was measured by the release of inorganic phosphate from radio-labelled $\gamma$-$^{32}$P-ATP in TLC. 10-μl reactions contained 5 mM $MgCl_2$, 0.01 mM ATP, 0.033 μM $\gamma$-$^{32}$P-ATP, and the indicated amounts of protein, either with or without 50 nM of an ssDNA substrate. Reactions were incubated at 37°C for 30 min and then stopped by addition of EDTA to a final concentration of 50 mM. 1 μl of the sample was spotted on a PEI-Cellulose TLC plate (Merck), which was placed into the mobile phase consisting of 0.15 M LiCl

and 0.15 M formic acid. After resolving, the plates were air-dried, wrapped in cling film and exposed to a storage phosphor screen (GE Healthcare). The signal was detected by scanning the screen with a Typhoon FLA9500 scanner (GE Healthcare).

## Primer extension assay

DDX11 variants (25 nM final) were preincubated with the primer–template substrate (20 nM final) in reaction buffer (10 mM Tris [pH 8.0], 25 mM KoAc, 0.5 mM DTT, and 0.1 mg/ml BSA) for 5 min on ice (mix A). Afterwards, an equal volume of mix B containing Pol δ (2.5 nM final), ATP (1 mM final), $MgCl_2$ (5 mM final), deoxynucleotide triphosphates (dNTPs) (0.1 mM each final) in reaction buffer was added, and the reactions (20 μl per time point) were immediately transferred to 37°C. At the indicated time points, 20 μl of the re-action were taken and added to a tube containing 20 μl 2× STOP solution (formamide containing 10 mM EDTA, 400 nM competitor, and bromophenol blue). The samples were boiled at 95°C for 7 min before loading on a 12% polyacrylamide gel (19:1 acrylamide to bis-acrylamide ratio) in 1× TBE containing 7 M urea. Gels were run at 18W for 2 h after a pre-run of 1 h at 10W. For signal detection, the gel was scanned with a Typhoon FLA9500 laser scanner (GE Healthcare) set to the fluorescence-imaging mode.

## Quantification of biochemical assays

Image quantification was performed in ImageJ (Schneider et al, 2012). For EMSAs, the percentage of substrate shifted relative to the total DNA (unshifted and shifted DNA) was quantified. For helicase assays, the signal intensity of the product band in the AMP-PNP control lane was subtracted as background. Then the percentage of product relative to the total DNA (substrate and product) was quantified. For ATPase assays, the signal intensity of the released inorganic phosphate (Pi) relative to the total signal (remaining ATP and released Pi) was quantified for all samples in the presence or absence of ssDNA. The background activity in the absence of DNA was then subtracted. For primer extension assays, the percentage of DNA at the block position (N+7 and N+8) relative to the total DNA (substrate only lane) was quantified. Bypass was then calculated by subtracting this value from 100.

## Mammalian cell culture

Human cells were cultured in a humidified incubator at 37°C and 6% $CO_2$ atmosphere in DMEM containing 4.5 g/l D-glucose, L-glutamine, and pyruvate (Gibco) supplemented with 5% FCS, except for RPE-1 that were supplemented with 10% FCS. To generate stable cell lines, HeLa Flp-In T-REx cells were transfected with Flp-In–compatible plasmids containing the gene of interest and an N-terminal YFP-tag (pDEST-YFP-FRT) along with a 10× excess of the pOG44 helper plasmid using Lipofectamine 2000 (Thermo Fisher Scientific) according to the manufacturer's instructions. To select for positive clones, a polyclonal selection was performed by diluting cells into selection medium containing 150 μg/ml hygromycin B and 15 μg/ml blasticidin (LabForce AG) and expanding resistant colonies.

**DNA substrates used in study.**

| Substrate | Oligo No. | Sequence (5′–3′) | Figure |
|---|---|---|---|
| 5′-overhang | 5′-FAM_42 | GACGCTGCCGAATTCTACCAGTGCCTTGCTAGGACATCTTTG | Figs 1C and E, 2C and E, and S1E |
|  | sAS13 | CAAAGATGTCCTAGCAAGGC |  |
| Competitor to 5′-overhang (helicase assay) | sAS14 | GCCTTGCTAGGACATCTTTG | Figs 1E and 2E |
| ssDNA | 5′-FAM_XO1 | GACGCTGCCGAATTCTACCAGTGCCTTGCTAGGACATCTTTGCCCACCTGCAGGTTCACCC | Figs 1D, 2D, and S1E |
| double-stranded DNA | 5′-FAM_XO1 | GACGCTGCCGAATTCTACCAGTGCCTTGCTAGGACATCTTTGCCCACCTGCAGGTTCACCC | Fig S1E |
|  | XO1c | GGGTGAACCTGCAGGTGGGCAAAGATGTCCTAGCAAGGCACTGGTAGAATTCGGCAGCGTC |  |
| Y-structure | 5′-FAM_42 | GACGCTGCCGAATTCTACCAGTGCCTTGCTAGGACATCTTTG | Fig S1E |
|  | sAS36 | CAAAGATGTCCTAGCAAGGCTTTTTTTTTTTTTTTTTTTTTT |  |
| 3′-overhang | 5′-FAM_42 | GACGCTGCCGAATTCTACCAGTGCCTTGCTAGGACATCTTTG | Fig S1E |
|  | sAS38 | TGGTAGAATTCGGCAGCGTC |  |
| Primer template | 5′-FAM_sAS50 | GGGTGAACCTGCAGGTGG | Fig 4B and C |
|  | XO1 | GACGCTGCCGAATTCTACCAGTGCCTTGCTAGGACATCTTTGCCCACCTGCAGGTTCACCC |  |
| DNA block | sAS51_3′-biotin | TGTCCTAGCAAGGCA |  |
| Competitor (primer extension assay) | XO1c | GGGTGAACCTGCAGGTGGGCAAAGATGTCCTAGCAAGGCACTGGTAGAATTCGGCAGCGTC |  |

### Drug treatment, plasmid transfections, and RNA interference

Indicated cells were treated with HU, aphidicolin, camptothecin, and/or the ATRi VE-821 (all Sigma-Aldrich). 293T cells were transiently transfected using the calcium phosphate transfection method. For RNA interference, the cells were transfected with siRNAs at a final concentration of 40 nM with DharmaFECT1 (GE Healthcare) and incubated for 48 h at 37°C.

### Cell extracts

Whole cell extracts were prepared by resuspending cell pellets in 4–5 pellet volumes of whole cell lysis buffer (50 mM sodium phosphate [pH 7.0], 150 mM NaCl, 0.5 mM EDTA, 1.5 mM $MgCl_2$, 10% Glycerol, 0.1% NP-40), supplemented freshly with cOmplete EDTA-free protease inhibitor cocktail (Roche), 1 mM DTT, and 0.1% benzonase (Santa Cruz). Cell extracts prepared for CHK1-pS345 detection were additionally treated with PhosSTOP phosphatase inhibitor (Roche). After a 20-min incubation on ice with occasional vortexing, the extracts were subjected to sonication for 2.5 min (high amplitude; 30 s ON and 30 s OFF) in a Bioruptor water bath sonicator (Diagenode) and centrifuged for 30 min at maximum speed in a table-top centrifuge.

### Cell fractionation

Cytoplasmic, nuclear, and chromatin fractions were obtained by sequential lysis of cell membranes using three buffers with varying amounts of detergent. All these buffers were supplemented freshly with cOmplete EDTA-free protease inhibitor cocktail (Roche). For cytoplasmic lysis, the cell pellets were carefully resuspended in 2.5 pellet volumes of cytoplasmic lysis buffer (150 mM NaCl, 10 mM Tris–HCl [pH 8.0], 1.5 mM $MgCl_2$, 10% glycerol, and 0.5% NP-40). After a 30-min incubation on ice, nuclei were pelleted by centrifugation for 2 min at 1,150$g$ in a table-top centrifuge, and the supernatant was kept as cytoplasmic fraction. Nuclei were resuspended in 2.5 pellet volumes of nuclear lysis buffer (150 mM NaCl, 20 mM Tris–HCl [pH 8.0], 1.5 mM $MgCl_2$, 1 mM EDTA, 10% glycerol, and 0.1% NP-40) and incubated for 1 h on ice. After a 30-min centrifugation at maximum speed, the nuclear fraction was recovered. The sedimented chromatin was in turn resuspended in 2.5 volumes of nuclease incubation buffer (150 mM NaCl, 20 mM Tris–HCl [pH 8.0], 1.5 mM $MgCl_2$, and 10% glycerol), supplemented freshly with 0.1% benzonase. Chromatin digestion was allowed to proceed on ice for 1 h. Then the chromatin fraction was subjected to sonication for 2.5 min (high amplitude; 30 s ON and 30 s OFF) and clarified by a 30-min centrifugation at maximum speed.

### Co-IP

To capture Flag-tagged proteins and their interaction partners, whole cell lysates were incubated overnight with ANTI-FLAG M2 Affinity beads (Sigma-Aldrich) or GFP-trap_A beads (ChromoTek). To remove unbound proteins, the beads were washed three times with lysis buffer. Finally, bound proteins were eluted from the beads by boiling in 2× protein loading buffer.

**siRNAs used in study.**

| siRNA | Sequence (5′–3′) | Figure |
|---|---|---|
| siControl | AGGUAGUGUAAUCGCCUUGtt | Used throughout the study |
| siDDX11 #1 | GGCGUUAGCUCCCGUAGGAtt | Used throughout the study |
| siDDX11 #2 | GAAUUCUGCCGGCGAAGAAtt | Used in Fig S4A |

## Mass spectrometry

N-terminally YFP-tagged DDX11 was expressed in stable HeLa Flp-In T-REx cells by doxycycline induction for 48 h. As a control, parental HeLa Flp-In T-REx cells were treated the same. Cell extracts were prepared by resuspending cell pellets in 4–5 pellet volumes of MS lysis buffer (10 mM Hepes [pH 7.9], 0.34 M sucrose, 3 mM $CaCl_2$, 2 mM magnesium acetate, 0.1 mM EDTA, 0.5% NP-40, and protease inhibitors). After a 10-min incubation on ice with occasional vortexing, the extracts were centrifuged for 20 min at 3,900$g$ in a table-top centrifuge. Cell lysates were then incubated with GFP-trap_A beads (ChromoTek) for 2 h at 4°C. The beads were washed three times for 20 min and bound proteins were eluted with two beads volumes of 0.2 M glycine (pH 2.5) and immediately neutralised with one fifth beads volume of 1M Tris base (pH 10.4).

Samples were analysed by mass spectrometry at the Functional Genomics Center Zurich. To do so, they were precipitated with an equal volume of 20% trichloroacetic acid (TCA; Sigma-Aldrich) and washed twice with cold acetone. The dry pellets were dissolved in 45 ml buffer (10 mM Tris [pH 8.2] and 2 mM $CaCl_2$) and 5 ml trypsin (100 ng/ml in 10 mM HCl) for digestion, which was carried out in a microwave instrument (Discover System; CEM) for 30 min at 5W and 60°C. The samples were dried in a SpeedVac (Savant). For liquid chromatography–tandem mass spectrometry (MS/MS) analysis, the samples were dissolved in 0.1% formic acid (Romil), and an aliquot ranging from 5 to 25% was analysed on a nanoAcquity UPLC System (Waters) connected to a QExactive mass spectrometer (Thermo Fisher Scientific) equipped with a Digital PicoView source (New Objective). Peptides were trapped on a Symmetry C18 trap column (5, 180 $\mu$m × 20 mm; Waters) and separated on a BEH300 C18 column (1.7, 75 $\mu$m × 150 mm; Waters) at a flow rate of 250 nl/min using a gradient from 1% solvent B (0.1% formic acid in acetonitrile; Romil)/99% solvent A (0.1% formic acid in water; Romil) to 40% solvent B/60% solvent A within 90 min.

The mass spectrometer was operated in data-dependent mode (DDA), acquiring a full-scan MS spectra (350–1,500 m/z) at a resolution of 70,000 at 200 m/z after accumulation to a target value of 3E6, followed by higher energy collision dissociation fragmentation on the 12 most intense signals per cycle. Higher energy collision dissociation spectra were acquired at a resolution of 35,000 using a normalized collision energy of 25 and a maximum injection time of 120 ms. The automatic gain control was set to 1E5 ions. Charge state screening was enabled and singly and unassigned charge states were rejected. Only precursors with intensity above 8,300 were selected for MS/MS (2% underfill ratio). Precursor masses previously selected for MS/MS measurement were excluded from further selection for 30 s, and the exclusion window was set at 10 ppm. The samples were acquired using internal lock mass calibration on m/z 371.1010 and 445.1200.

The MS raw-files were converted into Mascot generic files using ProteoWizard (http://proteowizard.sourceforge.net/). The peak lists were searched using Mascot Server (versions 2.4.1–2.6.2) against the Swiss-Prot human protein database. The search parameters were requirement for tryptic ends, two missed cleavage allowed, peptide tolerance ± 10 ppm, and MS/MS tolerance ± 0.03 kD. Oxidation (methionine) was set as variable modification. Scaffold (Proteome Software) was used to validate MS/MS-based

peptide and protein identifications. Peptide and protein identifications were accepted if they achieved a false discovery rate (calculated by Scaffold algorithm) of less than 0.1% and 1%, respectively. Only proteins identified with at least two peptides were considered for further analysis.

Gene Ontology terms analysis was performed using the PANTHER Classification System tool (http://www.pantherdb.org). The tool was fed with the list of proteins identified in both repeats. A statistical over-representation test with default settings was performed.

## Western blotting

Western blotting was performed following a standard protocol. Briefly, the samples were boiled in Laemmli sample buffer, separated by SDS–PAGE, and transferred onto nitrocellulose membranes. Afterwards, membranes were blocked in 5% milk-TBST (Tris-buffered saline supplemented with 0.01% Tween20) solution for 30 min and incubated with primary antibody solution overnight at 4°C. Then, the membranes were briefly washed with TBST and incubated with an appropriate secondary antibody (1:5,000) for 2 h at RT. The membranes were then washed several times with TBST, and the signal was developed using Clarity Western ECL Blotting Substrate (Bio-Rad) or SuperSignal West Femto Maximum Sensitivity Substrate (Thermo Fisher Scientific).

Image quantification was performed in ImageJ (Schneider et al, 2012). For cell fractionation experiments, the percentage of RPA on chromatin was calculated as the amount of RPA on chromatin relative to the amount of total RPA (cytoplasm + nucleoplasm + chromatin). For CHK1 activation experiments, the signal for CHK1-pS345 in siControl cells at 0 h was subtracted from the other signals as background. Then the ratio of CHK1-pS345 per total CHK1 was calculated for each time point. The value at the longest treatment point (2 or 4 h) in siControl cells was set to 100%, and all other values were calculated relative to this value.

## QIBC

RPE-1 cells seeded on coverslips were transfected with the indicated siRNAs using the DharmaFECT transfection reagent (GE Healthcare) and incubated for 24 h at 37°C. Then, 40 $\mu$M CldU were added, and the cells were incubated for an additional 24 h. After 0, 90, or 120 min of treatment with 2 mM HU and 2 $\mu$M ATRi (VE-821, SML1415; Sigma-Aldrich), the coverslips were transferred to a 24-well plate, and the cells were pre-extracted by incubation in 0.5 ml cold cytoskeletal (CSK) buffer (20 mM Hepes [pH 7.4], 50 mM NaCl, 300 mM sucrose, 3 mM $MgCl_2$, 1 mM EGTA, and 0.5% Triton X-100) on ice for 5 min. The cells were washed two times for 5 min with PBS and fixed in 0.5 ml fixing solution (4% paraformaldehyde in PBS, pH 7.2) for 12 min with gentle shaking. After one quick and one 5-min wash in PBS, fixed cells were incubated in 0.5 ml 0.3% Triton X-100/PBS for 10 min with gentle shaking. After two 5-min washes with PBS, unspecific binding sites were blocked by incubation in blocking solution (3% BSA in PBS) for 20 min. Afterwards, primary antibodies, diluted in blocking solution ($\alpha$-RPA34 [mouse, MA1-26418; Thermo Fisher Scientific]: 1:250; $\alpha$-BrdU [rat, ab6326; Abcam]: 1:250), were applied to the coverslips and incubated for 2 h in the dark. The

**Primary and secondary antibodies used for Western blots.**

| Protein/tag | Raised in | Provider | Catalogue no. |
|---|---|---|---|
| $\beta$-actin (C4)-HRP | Mouse | Santa Cruz | sc-47778 |
| CHK1 (pSer345) | Rabbit | Cell Signaling | 2348 |
| Total CHK1 | Mouse | Santa Cruz | sc-8408 |
| DDX11 | Mouse | Santa Cruz | sc-271711 |
| DNA Pol $\delta$ (p125) | Mouse | Abcam | ab196561 |
| FLAG M2 | Mouse | Sigma-Aldrich | F1804 |
| Histone H3 | Mouse | Abcam | ab10799 |
| MMS19 | Rabbit | Proteintech | 16015-1-AP |
| PCNA | Mouse | Santa Cruz | sc-56 |
| RPA34 | Mouse | Thermo Fisher Scientific | MA1-26418 |
| WDHD1 | Rabbit | Novus Biologicals | NBP1-89091 |
| Mouse IgG, HRP-linked | Sheep | GE Healthcare | NA931 |
| Rabbit IgG, HRP-linked | Donkey | GE Healthcare | NA934 |

coverslips were washed once with washing solution (0.05% Tween-20 in PBS) and once with PBS, before proceeding with the secondary antibody staining (Alexa488-conjugated $\alpha$-mouse [A-11001; Invitrogen] 1:500; Cy3-conjugated $\alpha$-rabbit [712-165-153; Jackson ImmunoResearch] 1:250 in Blocking solution). After a 45-min incubation, the coverslips were washed again once with washing solution and once with PBS. Nuclei were stained by incubation in 0.5 ml 0.5 $\mu$g/ml DAPI in PBS for 10 min. The coverslips were dipped three times in PBS and three times in MilliQ water, allowed to dry for a couple of minutes, and then mounted with 5 $\mu$l Mowiol-based mounting media (Mowiol 4.88 [Calbiochem] in Glycerol/TRIS). Automated multichannel wide-field microscopy for QIBC was performed on an Olympus ScanR Screening System equipped with an inverted motorized Olympus IX83 microscope, a motorized stage, IR-laser hardware autofocus, a fast emission filter wheel with single band emission filters, and a digital monochrome Hamamatsu ORCA-FLASH 4.0 V2 sCMOS camera (2,048 × 2,048 pixel, 12-bit dynamics). Images were acquired under non-saturating conditions at a single autofocus-directed z-position. Identical settings were applied to all samples within one experiment. Images were analysed with the inbuilt Olympus ScanR Image Analysis software version 3.0.1, a dynamic background correction was applied, and nuclei segmentation was performed using an integrated intensity-based object detection module based on the DAPI signal. All downstream analyses were focused on properly detected interphase nuclei containing a 2N-4N DNA content as measured by total and mean DAPI intensities. Fluorescence intensities were quantified and are depicted as arbitrary units. Colour-coded scatter plots of asynchronous cell populations were generated with Spotfire data visualisation software version 7.0.1 (TIBCO). Within one experiment, similar cell numbers were compared for the different conditions. Representative scatter plots containing several thousand cells per condition are shown.

## Supplementary Information

## Acknowledgements

We thank Stephen S Taylor for HeLa Flp-In T-REx cells, Robert M Brosh Jr for the pcDNA3-6xHis-DDX11-3xFlag plasmid, the Functional Genomics Center Zurich for mass spectrometry analyses, and the University of Zurich Center for Microscopy and Image Analysis for technical support. We are grateful to the entire Gari lab for helpful discussions. This project has received funding from the Swiss National Science Foundation (PP00P3_144784/1 and PP00P3_172959/1), the Human Frontier Science Program (CDA00043/2013-C), the "Stiftung für wissenschaftliche Forschung an der Universität Zürich," the Julius-Müller-Stiftung, the European Union's Horizon 2020 research and innovation programme under the Marie Skłodowska-Curie Grant Agreement No 707299, and the University of Zurich. AK Simon was recipient of a Candoc fellowship of the University of Zurich. A Lezaja, F Teloni, and M Altmeyer received funding from the Swiss National Science Foundation (PP00P3_179057) and the European Research Council (ERC) under the European Union's Horizon 2020 research and innovation program (ERC-StG 714326).

### Author Contributions

AK Simon: conceptualization, formal analysis, funding acquisition, investigation, visualization, and writing—review and editing.
S Kummer: formal analysis and investigation.
S Wild: formal analysis, investigation, and writing—review and editing.
A Lezaja: formal analysis, investigation, visualization, and writing—review and editing.
F Teloni: formal analysis, investigation, visualization, and writing—review and editing.
SK Jozwiakowski: resources and writing—review and editing.
M Altmeyer: formal analysis, supervision, funding acquisition, visualization, and writing—review and editing.
K Gari: conceptualization, formal analysis, supervision, funding acquisition, validation, investigation, visualization, project administration, and writing—original draft, review, and editing.

### Conflict of Interest Statement

The authors declare that they have no conflict of interest.

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
