## [Reviewer comments · Life Science Alliance]

Life Science Alliance

The iron-sulfur helicase DDX11 promotes the generation of single-stranded DNA for CHK1 activation

Anna Simon, Sandra Kummer, Sebastian Wild, Aleksandra Lezaja, Federico Teloni, Stanislaw Jozwiakowski, Matthias Altmeyer, and Kerstin Gari

DOI: <https://doi.org/10.26508/lsa.201900547>

Corresponding author(s): Kerstin Gari, University of Zurich

Review Timeline:

Submission Date:	2019-09-11
Editorial Decision:	2019-10-03
Revision Received:	2020-01-29
Editorial Decision:	2020-02-03
Revision Received:	2020-02-06
Accepted:	2020-02-06

Scientific Editor: Andrea Leibfried

Transaction Report:

October 3, 2019

Re: Life Science Alliance manuscript #LSA-2019-00547-T

Prof. Kerstin Gari
University of Zurich
Institute of Molecular Cancer Research
Winterthurerstrasse 190
Zurich, Zurich 8057
Switzerland

Dear Dr. Gari,

Thank you for submitting your manuscript entitled "The iron-sulfur helicase DDX11 promotes the generation of single-stranded DNA for CHK1 signalling" to Life Science Alliance. The manuscript was assessed by expert reviewers, whose comments are appended to this letter.

As you will see, all three reviewers are supportive of publication here, pending revision. We would thus like to invite you to submit a revised version of your manuscript to us, addressing the criticisms raised by reviewer #2 and #3. The work needs to get better placed into the existing literature and the mutant analyses should get extended and missing controls added. Reviewer #2 also thinks that additional support for the conclusion that DDX11 depletion prevents ssDNA accumulation and checkpoint signaling is needed, and we agree that it would be good to add such support.

Thank you for this interesting contribution to Life Science Alliance. We are looking forward to receiving your revised manuscript.

Sincerely,

B. MANUSCRIPT ORGANIZATION AND FORMATTING:

Reviewer #1 (Comments to the Authors (Required)):

This excellent report by Simon et al. that describes the importance of the iron sulphur cluster in

DDX11. Mutations in DDX11 are associated with Warsaw Breakage Syndrome.

In figure 1 they show that mutations in the iron sulphur cluster that prevent iron binding are associated with loss of function in biochemical assays. The protein is essentially inactive.

In figure 2 they show that replacement of arginine-263 with positively charged amino acids that can still coordinate iron partially restores function compared to a neutral amino acid. Activity thus correlates with ability to coordinate iron.

In figure 3 they show that DDX11 binds to proteins associated with the replisome including WDHD1 and Pol Delta. Reciprocal co-IPs are shown. The authors use benzonase to prove that the interaction is direct rather than DNA-mediated.

In figure 4 they show that wild-type DDX11 is required to remove obstacles in a primer extension assay.

IN figures 5 and 6 they show that DDX11 is required for the generation of ssDNA and CHK1 DNA damage signaling after HU and aphidicholin.

This is an excellent paper.

Reviewer #2 (Comments to the Authors (Required)):

In their manuscript, Simon et al describe the contribution of the iron-sulfur cluster in the DDX1 helicase to its various biochemical functions. The authors then claim that DDX1 promotes checkpoint signaling as a consequence of its ability to dismantle secondary structures in DNA. I believe that the in vitro data are clear and convincing and provide useful information to the field. However, I'm afraid that the in vivo data are fragmented and not convincing. Moreover, the data are not appropriately discussed in the context of previous literature; indeed many in vivo data are not novel. Therefore, I think that the second part of the manuscript needs to be revised and improved. Regarding the presentation of the data, I did not find specified how many times each experiment was performed. Besides, some experiments lack important controls (see below). Also, statistical analysis is missing throughout the manuscript.

Major issues:

1. The Co-IPs shown in Figs 3 and S3 were done using whole-cell extracts and over-expressed proteins. Have the authors tried to work with endogenous proteins, chromatin fractions or at least S phase cells? What is the novelty or the point that the authors want to make with the Co-IPs? As the authors themselves point out, DDX1 has already been shown to interact with factors involved in lagging strand replication. Moreover, DDX1 has been shown to localize to the replication fork (ref 14 in the manuscript). Also in this regard, how do the authors explain that DDX1 is not detected in the chromatin fraction (Fig 5C)? I think that Fig 3 should focus on the pold1-ddx1 interaction (at chromatin or forks) in control and e.g. HU/ATRi conditions.

2. Figs 5 and 6 show that DDX1 depletion causes reduced RPA chromatin binding and reduced Chk1 phosphorylation. There are no controls showing S phase levels in control and DDX1-depleted cells; e.g. do control and DDX1-depleted cells incorporate CldU at the same rate? It would also be important to include representative images, in particular of ssDNA, and it would be interesting to determine and compare it to gH2AX levels. In Figs 5A-B the controls without HU/ATRi are missing and in Fig 5C control and HU/ATRi are shown as separate blots. Given the authors claims, the difference expected between control and HU/ATRi should be exacerbated by DDX1 depletion.

3. The authors imply that the FeS cluster in DDX1 and its helicase function are required for DDX11-dependent ssDNA formation. Thus, the experiments in Fig 5 should be repeated including the DDX1 mutants.

4. A major claim made by the authors is that DDX1 promotes checkpoint signaling. This is only based on the western blots in Fig 6 showing Chk1 phosphorylation. First, the western blot should be quantified, because it is clear that not only Chk1-ps345, but also Chk1 levels, are reduced upon DDX1 depletion. Second, other outputs of checkpoint signaling should be analyzed, e. g. origin firing levels, which are indeed induced upon Chk1 depletion (Cali, NAR, 2016). Otherwise, the authors should tone down their claims and talk only about Chk1 phosphorylation. The DDX1 mutants should be included in these experiments, including the western blots in Fig. 6.

5. To confirm that DDX1 depletion prevents ssDNA accumulation (and in turn checkpoint signaling) instead of DNA damage accumulation, additional experiments should be done. As already mentioned in point 2, gH2AX levels should be determined. Other possibilities include sister chromatid cohesion or survival assays. Moreover, I think the manuscript would greatly benefit from including DNA fiber assays. In agreement with the authors' model, Cali et al show that DDX1 depletion reduces DNA elongation upon HU treatment. It would be interesting to check if the mutants complement this phenotype, and this would serve to support or redefine the model proposed by the authors.

6. The authors propose a model in which DDX1 localizes at the lagging strand to dismantle DNA secondary structures, thereby generating ssDNA and promoting checkpoint signaling. The effect of DDX1 depletion on checkpoint signaling has already been reported in HeLa cells (Cali, NAR, 2016). However, this paper is not cited. Moreover, a possible function of DDX1 in the unwinding of secondary structures has already been proposed (e.g. Pisani, Genes, 2018). This is not clearly stated in the manuscript and the authors even write: "given ddx11's interactions with lagging-strand proteins and its helicase activity, we reasoned that one possible role of DDX1 could be the removal of secondary structures during DNA replication". These kind of sentences should be rewritten and the appropriate references included. This will also help clarify what is the specific contribution of this paper to the field.

Briefly, I consider unavoidable to include the mutants in Figs 5 and 6, include the missing controls and improve the writing, adding the corresponding references where needed. Also, the conclusions should be further supported by additional experiments like the DNA fiber assay and measurement of biologically relevant parameters.

Minor issues:

1. Primers used for mutagenesis are not listed and some specifications are missing, for example that cysteines were substituted by serines or what kind of mutant is K50R and why it is used.
2. Fig 2: I do not understand why the authors choose to show the EMSA and the ATP assay in the supplement. I would move the data to the main figure, so as to have it symmetrical with Fig 1. More importantly, I think it might be misleading not to show the EMSA in Fig 2, as it can be interpreted as the mutant being specifically impaired in helicase activity, whereas it is its DNA binding activity that is impaired, and, as a consequence, the helicase function.
3. Fig 4: it might be useful to add "bp" (base pairs) next to N+7 and also next to the numbers in panel A.
4. Fig S3A: Flag-DDX1 is not present in the input
5. Fig S4: Controls as those shown in Fig 4 are missing.

Reviewer #3 (Comments to the Authors (Required)):

The authors show that a mutation in the arginine-263 of the FeS cluster of DDX11 negatively

impacts on the the DNA binding, ATP hydrolysis and DNA helicase activity of DDX11. Using IP-mass spectrometry followed by co-IP, the authors go on to show that DDX11 interacts with several replisome components, among which DNA polymerase delta and Ctf4/WDHD1. Using in vitro assays, the authors show that DDX11 can unwind DNA substrates containing DNA blocks generating a single stranded substrate for Pol delta. This activity of DDX11 relies on its ATPase and FeS domain. To match these results with the in vivo function of DDX11, the authors address if checkpoint signalling upon treatment with drugs inhibiting replication fork progression is dependent on DDX11. They find evidence for reduced exposure of ssDNA coated with RPA that results in defective checkpoint signalling.

The results are convincing and the data is solid. Overall the study is of interest and reports novel and interesting results. Therefore, this study deserves to be published in Life Science Alliance. I would like to suggest some minor textual modifications that will place the new study in the context of previous work. For instance, previous work in budding yeast and DT40 cells did not find a prominent role for Chl1/DDX11 in checkpoint signalling (Laha et al, NAR, 2006; Abe et al, PNAS, 2018). Moreover, previous work identified strong interactions between DDX11 and Ctf18-RFC and Fen1, but not with Ctf4 (Farina et al, JBC, 2008). It will be useful for readers to be aware of these differences even if the exact reasons underlying these discrepancies are not obvious now.

Reviewer #1 (Comments to the Authors (Required)):

This excellent report by Simon et al. that describes the importance of the iron sulphur cluster in DDX11. Mutations in DDX11 are associated with Warsaw Breakage Syndrome.

In figure 1 they show that mutations in the iron sulphur cluster that prevent iron binding are associated with loss of function in biochemical assays. The protein is essentially inactive.

In figure 2 they show that replacement of arginine-263 with positively charged amino acids that can still coordinate iron partially restores function compared to a neutral amino acid. Activity thus correlates with ability to coordinate iron.

In figure 3 they show that DDX11 binds to proteins associated with the replisome including WDHD1 and Pol Delta. Reciprocal co-IPs are shown. The authors use benzonase to prove that the interaction is direct rather than DNA-mediated.

In figure 4 they show that wild-type DDX11 is required to remove obstacles in a primer extension assay.

IN figures 5 and 6 they show that DDX11 is required for the generation of ssDNA and CHK1 DNA damage signaling after HU and aphidicholin.

This is an excellent paper.

We thank the reviewer for this very positive feedback.

Reviewer #2 (Comments to the Authors (Required)):

In their manuscript, Simon et al describe the contribution of the iron-sulfur cluster in the DDX1 helicase to its various biochemical functions. The authors then claim that DDX1 promotes checkpoint signaling as a consequence of its ability to dismantle secondary structures in DNA. I believe that the in vitro data are clear and convincing and provide useful information to the field. However, I'm afraid that the in vivo data are fragmented and not convincing. Moreover, the data are not appropriately discussed in the context of previous literature; indeed many in vivo data are not novel. Therefore, I think that the second part of the manuscript needs to be revised and improved.

Regarding the presentation of the data, I did not find specified how many times each experiment was performed. Besides, some experiments lack important controls (see below). Also, statistical analysis is missing throughout the manuscript.

We thank the reviewer for their feedback. In the course of the revision, we have included new experiments. Moreover, we have added quantifications, statistical analyses and information on biological replicates. We have also made sure to discuss the literature more appropriately.

Major issues:

1. The Co-IPs shown in Figs 3 and S3 were done using whole-cell extracts and over-expressed proteins. Have the authors tried to work with endogenous proteins, chromatin fractions or at least S phase cells? What is the novelty or the point that the authors want to make with the Co-IPs? As the authors themselves point out, DDX1 has already been shown to interact with factors involved in

lagging strand replication. Moreover, DDX1 has been shown to localize to the replication fork (ref 14 in the manuscript). Also in this regard, how do the authors explain that DDX1 is not detected in the chromatin fraction (Fig 5C)? I think that Fig 3 should focus on the pold1-ddx1 interaction (at chromatin or forks) in control and e.g. HU/ATRi conditions.

Using mass spectrometry, we find endogenous POLD1 with YFP-tagged DDX11 (Figure 3A) and endogenous DDX11 with Flag-tagged POLD1 (see **attachment 1**). We can also detect endogenous POLD1 by Western blot (Figure 3B) when performing a pull-down experiment with Flag-tagged DDX11. However, we were not able to pull down endogenous POLD1 with endogenous DDX11 or *vice versa*, possibly because this approach is not sensitive enough.

Given our finding that DDX11 functionally interacts with DNA Pol delta *in vitro* (Figure 4B,C), we consider it noteworthy that DDX11 and POLD1 are also able to interact. We agree with the reviewer that the observed interaction between DDX11 and WDHD1 is less directly relevant to our story, but do nevertheless consider it an important information for the more specialized audience. We have explained our motivation to study these interactions better and have concentrated in Figure 3 on the interaction of DDX11 with POLD1, while moving the WDHD1 part to Figure S3.

As for the localization of DDX11 – we agree that it is puzzling that not more DDX11 is found on chromatin (not even after replication stress treatment). However, we have repeated this experiment several times, also with a different fractionation protocol (see **attachment 2**), and have always obtained the same result. DDX11 might reside in the cytoplasm and nucleoplasm most of the time and only very transiently localize to chromatin. It is also noteworthy that the interaction of DDX11 with POLD1 does not seem to increase with HU treatment (Table S1).

2. Figs 5 and 6 show that DDX1 depletion causes reduced RPA chromatin binding and reduced Chk1 phosphorylation. There are no controls showing S phase levels in control and DDX1-depleted cells; e.g. do control and DDX1-depleted cells incorporate CldU at the same rate? It would also be important to include representative images, in particular of ssDNA, and it would be interesting to determine and compare it to γ H2AX levels. In Figs 5A-B the controls without HU/ATRi are missing and in Fig 5C control and HU/ATRi are shown as separate blots. Given the authors claims, the difference expected between control and HU/ATRi should be exacerbated by DDX1 depletion.

We now show control graphs (non-treated conditions) for the QIBC experiment (Figures 5A,B). They demonstrate that CldU is incorporated at a similar rate in DDX11-depleted cells and control cells. We now also include representative images showing CldU and RPA staining in randomly selected S-phase cells (Figure 5C).

As for γ H2AX – we now include Western blot analysis of γ H2AX in response to HU, Aph and CPT (Figure 6), which shows no difference between DDX11-depleted and control cells.

We have repeated the cell fractionation experiment to have all fractions on one gel and have included quantifications of the amount of RPA on chromatin (Figure 5D).

3. The authors imply that the FeS cluster in DDX1 and its helicase function are required for DDX11-dependent ssDNA formation. Thus, the experiments in Fig 5 should be repeated including the DDX1 mutants.

This is a very good point raised by the reviewer. Unfortunately, RPE1 cells are very difficult to transfect. We have spent a considerable amount of time trying to optimize the transfection conditions, but could not achieve a good transfection efficiency with various transfection reagents (Fugene, Lipofectamine 2000, branched PEI). We had a good transfection efficiency with Jetprime, but observed a very robust CHK1 phosphorylation through the transfection reagent itself (see **attachment 3**).

We also tried U2OS and HeLa cells that are more amenable to plasmid transfections. However, while we could observe a reduced CHK1 phosphorylation upon *DDX11* knock-down, the effect was less pronounced, which did not allow a complementation analysis (see **attachment 4**). This is in line with previous data that showed a slight reduction in CHK1-pS345 upon *DDX11* knock-down in HeLa cells (Figure S5 in Cali, NAR, 2016).

We have re-phrased the relevant section in the discussion to make clear that we cannot experimentally prove that the observed effect on ssDNA is due to *DDX11*'s FeS cluster and/or helicase activity:

To exaggerate the formation of replication-dependent secondary structures in vivo and to address whether DDX11 could play a role in their resolution, we treated cells with HU and ATRi. In the absence of DDX11, such a treatment led to reduced levels of native ssDNA and chromatin-bound RPA. While these findings are in agreement with a role for DDX11 in the unwinding of DNA secondary structures, future studies will have to demonstrate whether DDX11's helicase activity or an intact FeS domain are indeed required for this function.

We have also included a second siRNA to rule out that the effect on CHK1-pS345 is caused by siRNA off-target effects (Figure S4A).

4. A major claim made by the authors is that *DDX1* promotes checkpoint signaling. This is only based on the western blots in Fig 6 showing Chk1 phosphorylation. First, the western blot should be quantified, because it is clear that not only Chk1-pS345, but also Chk1 levels, are reduced upon *DDX1* depletion. Second, other outputs of checkpoint signaling should be analyzed, e. g. origin firing levels, which are indeed induced upon Chk1 depletion (Cali, NAR, 2016). Otherwise, the authors should tone down their claims and talk only about Chk1 phosphorylation. The *DDX1* mutants should be included in these experiments, including the western blots in Fig. 6.

The reviewer is absolutely right that also the levels of total CHK1 are affected by depletion of *DDX11*, albeit to a lesser degree. In the revised manuscript, we provide the quantification of CHK1-pS345 and total CHK1 levels (from three independent experiments) and a tentative explanation for this observation:

Interestingly, also the levels of total CHK1 were reduced by depletion of DDX11, even in untreated (0h) conditions (Figures 6A–C). This may be explained by recent work establishing the need for a basal CHK1 activity during unchallenged conditions to prevent CHK1 degradation (Michelena et al, 2019). We suspect that DDX11 may be required for CHK1's steady-state activity, and that loss of DDX11 may in turn affect CHK1 stability.

We also make sure to be careful with our wording and to integrate our data more appropriately in the existing literature:

In line with our observation that DDX11 promotes the generation of ssDNA and the loading of RPA on chromatin, phosphorylation of the ATR effector kinase CHK1 at serine-345 was greatly impaired upon DDX11 depletion in response to a variety of replication stress-inducing agents, suggesting a role for DDX11 in CHK1 activation. These findings are in agreement with a previous study, in which knock-down of DDX11 – in addition to a slight reduction in CHK1-pS345 – caused impaired fork restart and increased origin firing upon HU treatment (Cali et al, 2016).

As discussed above, we could unfortunately not set up a complementation system to analyse the contribution of the FeS cluster to *DDX11*'s role in fork dynamics.

5. To confirm that DDX1 depletion prevents ssDNA accumulation (and in turn checkpoint signaling) instead of DNA damage accumulation, additional experiments should be done. As already mentioned in point 2, γ H2AX levels should be determined. Other possibilities include sister chromatid cohesion or survival assays. Moreover, I think the manuscript would greatly benefit from including DNA fiber assays. In agreement with the authors' model, Cali et al show that DDX1 depletion reduces DNA elongation upon HU treatment. It would be interesting to check if the mutants complement this phenotype, and this would serve to support or redefine the model proposed by the authors.

To check whether depletion of DDX11 leads to the accumulation of DNA damage, we now also show γ H2AX levels, which display no apparent difference between DDX11-depleted and control cells. Again, we could unfortunately not carry out any complementation analysis.

6. The authors propose a model in which DDX1 localizes at the lagging strand to dismantle DNA secondary structures, thereby generating ssDNA and promoting checkpoint signaling. The effect of DDX1 depletion on checkpoint signaling has already been reported in HeLa cells (Cali, NAR, 2016). However, this paper is not cited. Moreover, a possible function of DDX1 in the unwinding of secondary structures has already been proposed (e.g. Pisani, Genes, 2018). This is not clearly stated in the manuscript and the authors even write: "given ddx11's interactions with lagging-strand proteins and its helicase activity, we reasoned that one possible role of DDX1 could be the removal of secondary structures during DNA replication". These kind of sentences should be rewritten and the appropriate references included. This will also help clarify what is the specific contribution of this paper to the field.

We are aware that DDX11 – being a helicase found close to replication forks – has already been suggested to play a role in the removal of secondary structures during DNA replication (e.g. in the review by Pisani et al., Genes, 2018). To our knowledge, however, we provide a first direct evidence for DDX11 being able to remove secondary structures ahead of Pol delta. That being said, we have made sure to discuss the literature more appropriately:

Given DDX11's interactions with lagging-strand proteins and its helicase activity, a role for DDX11 in the resolution of secondary structures during DNA replication has been discussed (Bharti et al, 2014; Pisani et al, 2018). To directly address whether DDX11 may be able to functionally interact with Pol delta, e.g. by removing DNA obstacles from its DNA template, we performed time-resolved primer extension assays with purified Pol delta and DDX11 variants (Figure 4).

We now also cite the work by Cali et al. (see point 4).

Briefly, I consider unavoidable to include the mutants in Figs 5 and 6, include the missing controls and improve the writing, adding the corresponding references where needed. Also, the conclusions should be further supported by additional experiments like the DNA fiber assay and measurement of biologically relevant parameters.

In the revised manuscript, we have included all missing controls and made sure to integrate our data more appropriately in the existing literature. We have also repeated the CHK1 activation assays and checked for the induction of DNA damage (γ H2AX signal) in the absence of DDX11. Due to the lack of a depletion/complementation system, we could, however, unfortunately not check the contribution of the FeS cluster and the helicase domain.

Minor issues:

1. Primers used for mutagenesis are not listed and some specifications are missing, for example that cysteines were substituted by serines or what kind of mutant is K50R and why it is used.

We have added the missing information on the mutagenesis primers and explained better the DDX11 variants used throughout the study.

2. Fig 2: I do not understand why the authors choose to show the EMSA and the ATP assay in the supplement. I would move the data to the main figure, so as to have it symmetrical with Fig 1. More importantly, I think it might be misleading not to show the EMSA in Fig 2, as it can be interpreted as the mutant being specifically impaired in helicase activity, whereas it is its DNA binding activity that is impaired, and, as a consequence, the helicase function.

We have moved the EMSA and ATPase assay from Fig. S2 to Fig. 2.

3. Fig 4: it might be useful to add "bp" (base pairs) next to N+7 and also next to the numbers in panel A.

We have amended the figure legend to make clear what the numbers mean.

4. Fig S3A: Flag-DDX1 is not present in the input

This was a detection problem due to a not very well working anti-Flag antibody batch. We have repeated these co-IP experiments (now Fig. 3D) and used a DDX11 antibody (instead of an anti-Flag antibody) to detect Flag-DDX11 in the input fractions.

5. Fig S4: Controls as those shown in Fig 4 are missing.

The experiments in Figures 4 and S4 were done in parallel. We have therefore decided to show them as one figure (Figure 4).

Reviewer #3 (Comments to the Authors (Required)):

The authors show that a mutation in the arginine-263 of the FeS cluster of DDX11 negatively impacts on the the DNA binding, ATP hydrolysis and DNA helicase activity of DDX11. Using IP-mass spectrometry followed by co-IP, the authors go on to show that DDX11 interacts with several replisome components, among which DNA polymerase delta and Ctf4/WDHD1. Using in vitro assays, the authors show that DDX11 can unwind DNA substrates containing DNA blocks generating a single stranded substrate for Pol delta. This activity of DDX11 relies on its ATPase and FeS domain. To match these results with the in vivo function of DDX11, the authors address if checkpoint signalling upon treatment with drugs inhibiting replication fork progression is dependent on DDX11. They find evidence for reduced exposure of ssDNA coated with RPA that results in defective checkpoint signalling.

The results are convincing and the data is solid. Overall the study is of interest and reports novel and

interesting results. Therefore, this study deserves to be published in Life Science Alliance. I would like to suggest some minor textual modifications that will place the new study in the context of previous work. For instance, previous work in budding yeast and DT40 cells did not find a prominent role for Chl1/DDX11 in checkpoint signalling (Laha et al, NAR, 2006; Abe et al, PNAS, 2018). Moreover, previous work identified strong interactions between DDX11 and Ctf18-RFC and Fen1, but not with Ctf4 (Farina et al, JBC, 2008). It will be useful for readers to be aware of these differences even if the exact reasons underlying these discrepancies are not obvious now.

We thank the reviewer for their feedback. We have made sure to discuss the above-mentioned papers more appropriately:

Surprisingly, however, the function of DDX11 in CHK1 activation does not seem to be conserved in budding yeast and DT40 cells where phosphorylation of CHK1 was unaffected by the absence of DDX11 (Laha et al, 2006; Abe et al, 2018).

Since a previous study made mention of only a weak interaction between human DDX11 and CTF4/WDHD1 (Farina et al, 2008), and no obvious CIP box motif is found in human DDX11, it has however remained unclear whether the interaction between DDX11 and WDHD1 is conserved in humans.

	Identified Proteins with Flag-POLD1 (293T cells)	Accession Number	MW	control	Flag PD
1	DNA polymerase delta catalytic subunit OS=Homo sapiens GN=POLD1 PE=1 SV=2	DPOD1_HUMAN	124 kDa	0	89
2	T-complex protein 1 subunit beta OS=Homo sapiens GN=CCT2 PE=1 SV=4	TCPB_HUMAN	57 kDa	0	29
3	Nucleolar RNA helicase 2 OS=Homo sapiens GN=DDX21 PE=1 SV=5	DDX21_HUMAN	87 kDa	0	28
4	T-complex protein 1 subunit gamma OS=Homo sapiens GN=CCT3 PE=1 SV=4	TCPG_HUMAN	61 kDa	0	28
5	T-complex protein 1 subunit theta OS=Homo sapiens GN=CCT8 PE=1 SV=4	TCPQ_HUMAN	60 kDa	0	27
6	ATP-dependent RNA helicase A OS=Homo sapiens GN=DHX9 PE=1 SV=4	DHX9_HUMAN	141 kDa	0	26
7	T-complex protein 1 subunit eta OS=Homo sapiens GN=CCT7 PE=1 SV=2	TCPH_HUMAN	59 kDa	0	26
8	Nuclear mitotic apparatus protein 1 OS=Homo sapiens GN=NUMA1 PE=1 SV=2	NUMA1_HUMAN	238 kDa	0	23
9	T-complex protein 1 subunit delta OS=Homo sapiens GN=CCT4 PE=1 SV=4	TCPD_HUMAN	58 kDa	0	23
10	Myb-binding protein 1A OS=Homo sapiens GN=MYBBP1A PE=1 SV=2	MBB1A_HUMAN	149 kDa	0	22
11	Stress-induced-phosphoprotein 1 OS=Homo sapiens GN=STIP1 PE=1 SV=1	STIP1_HUMAN	63 kDa	0	22
12	T-complex protein 1 subunit alpha OS=Homo sapiens GN=TCP1 PE=1 SV=1	TCPA_HUMAN	60 kDa	0	21
13	Nucleolin OS=Homo sapiens GN=NCL PE=1 SV=3	NUCL_HUMAN	77 kDa	0	19
14	T-complex protein 1 subunit zeta OS=Homo sapiens GN=CCT6A PE=1 SV=3	TCPZ_HUMAN	58 kDa	0	19
15	Heat shock protein 105 kDa OS=Homo sapiens GN=HSPH1 PE=1 SV=1	HS105_HUMAN	97 kDa	0	18
16	40S ribosomal protein S3 OS=Homo sapiens GN=RPS3 PE=1 SV=2	RS3_HUMAN	27 kDa	0	16
17	BAG family molecular chaperone regulator 2 OS=Homo sapiens GN=BAG2 PE=1 SV=1	BAG2_HUMAN	24 kDa	0	15
18	BAG family molecular chaperone regulator 5 OS=Homo sapiens GN=BAG5 PE=1 SV=1	BAG5_HUMAN	51 kDa	0	15
19	Pre-mRNA-processing-splicing factor 8 OS=Homo sapiens GN=PRPF8 PE=1 SV=2	PRP8_HUMAN	274 kDa	0	14
20	DNA polymerase delta subunit 3 OS=Homo sapiens GN=POLD3 PE=1 SV=2	DPOD3_HUMAN	51 kDa	0	14
21	60S ribosomal protein L5 OS=Homo sapiens GN=RPL5 PE=1 SV=3	RL5_HUMAN	34 kDa	0	13
22	DNA polymerase delta subunit 2 OS=Homo sapiens GN=POLD2 PE=1 SV=1	DPOD2_HUMAN	51 kDa	0	12
23	Putative ATP-dependent RNA helicase DHX30 OS=Homo sapiens GN=DHX30 PE=1 SV=1	DHX30_HUMAN	134 kDa	0	11
24	40S ribosomal protein S13 OS=Homo sapiens GN=RPS13 PE=1 SV=2	RS13_HUMAN	17 kDa	0	11
25	Bystin OS=Homo sapiens GN=BYSL PE=1 SV=3	BYST_HUMAN	50 kDa	0	11
26	Heat shock 70 kDa protein 4 OS=Homo sapiens GN=HSPA4 PE=1 SV=4	HSP74_HUMAN	94 kDa	0	11
27	RNA cytidine acetyltransferase OS=Homo sapiens GN=NAT10 PE=1 SV=2	NAT10_HUMAN	116 kDa	0	11
28	Cluster of Histone H1.2 OS=Homo sapiens GN=HIST1H1C PE=1 SV=2 (H12_HUMAN)	H12_HUMAN [3]	21 kDa	0	10
29	40S ribosomal protein S19 OS=Homo sapiens GN=RPS19 PE=1 SV=2	RS19_HUMAN	16 kDa	0	10
30	Ribosomal L1 domain-containing protein 1 OS=Homo sapiens GN=RSL1D1 PE=1 SV=3	RL1D1_HUMAN	55 kDa	0	10
31	Probable 28S rRNA (cytosine(4447)-C(5))-methyltransferase OS=Homo sapiens GN=NOP2 PE=1 SV=2	NOP2_HUMAN	89 kDa	0	10
32	Transcription intermediary factor 1-beta OS=Homo sapiens GN=TRIM28 PE=1 SV=5	TIF1B_HUMAN	89 kDa	0	10
33	Elongation factor Tu, mitochondrial OS=Homo sapiens GN=TUFM PE=1 SV=2	EFTU_HUMAN	50 kDa	0	10
34	Matrin-3 OS=Homo sapiens GN=MATR3 PE=1 SV=2	MATR3_HUMAN	95 kDa	0	10
35	Eukaryotic initiation factor 4A-I OS=Homo sapiens GN=EIF4A1 PE=1 SV=1	IF4A1_HUMAN	46 kDa	0	10
36	40S ribosomal protein S5 OS=Homo sapiens GN=RPS5 PE=1 SV=4	RS5_HUMAN	23 kDa	0	9
37	60S ribosomal protein L9 OS=Homo sapiens GN=RPL9 PE=1 SV=1	RL9_HUMAN	22 kDa	0	9
38	Cluster of ATP-dependent DNA helicase DDX11 OS=Homo sapiens GN=DDX11 PE=1 SV=1 (DDX11_HUMAN)	DDX11_HUMAN	108 kDa	0	9
39	60S ribosomal protein L32 OS=Homo sapiens GN=RPL32 PE=1 SV=2	RL32_HUMAN	16 kDa	0	9
40	60S ribosomal protein L28 OS=Homo sapiens GN=RPL28 PE=1 SV=3	RL28_HUMAN	16 kDa	0	9
41	WD repeat-containing protein 6 OS=Homo sapiens GN=WDR6 PE=1 SV=1	WDR6_HUMAN	122 kDa	0	9
42	DnaJ homolog subfamily A member 1 OS=Homo sapiens GN=DNAJA1 PE=1 SV=2	DNJA1_HUMAN	45 kDa	0	9
43	rRNA 2'-O-methyltransferase fibrillarin OS=Homo sapiens GN=FBRL PE=1 SV=2	FBRL_HUMAN	34 kDa	0	9
44	40S ribosomal protein S7 OS=Homo sapiens GN=RPS7 PE=1 SV=1	RS7_HUMAN	22 kDa	0	9
45	Insulin-like growth factor 2 mRNA-binding protein 1 OS=Homo sapiens GN=IGF2BP1 PE=1 SV=2	IF2B1_HUMAN	63 kDa	0	9
46	E3 ubiquitin-protein ligase CHIP OS=Homo sapiens GN=STUB1 PE=1 SV=2	CHIP_HUMAN	35 kDa	0	9
47	Serine/arginine repetitive matrix protein 2 OS=Homo sapiens GN=SRRM2 PE=1 SV=2	SRRM2_HUMAN	300 kDa	0	8
48	Tyrosine-protein kinase BAZ1B OS=Homo sapiens GN=BAZ1B PE=1 SV=2	BAZ1B_HUMAN	171 kDa	0	8
49	60S ribosomal protein L13 OS=Homo sapiens GN=RPL13 PE=1 SV=4	RL13_HUMAN	24 kDa	0	8
50	60S ribosomal protein L17 OS=Homo sapiens GN=RPL17 PE=1 SV=3	RL17_HUMAN	21 kDa	0	8

Attachment 1.
Top 50 hits identified by mass spectrometry with Flag-tagged POLD1 pulled-down from 293T cells.

Attachment 2.

Localisation of DDX11 using a different fractionation protocol.

Attachment 3.

Transfection of YFP-tagged DDX11 variants (wt, K50R, C267S, R263Q) or GFP alone in RPE-1 cells using Jetprime. Compare CHK1-pS345 of lane 2 (Jetprime) and lane 7 (no transfection reagent) – both samples are efficiently depleted of *DDX11* (lower band).

A**B****Attachment 4.**

(A) Western blots showing time course of CHK1-pS345 activation in control U2OS cells (siControl) and cells depleted of *DDX11* (siDDX11). (B) Western blots showing time course of CHK1-pS345 activation in control HeLa cells (siControl) and cells depleted of *DDX11* (siDDX11).

February 3, 2020

RE: Life Science Alliance Manuscript #LSA-2019-00547-TR

Dr. Kerstin Gari
University of Zurich
Institute of Molecular Cancer Research
Winterthurerstrasse 190
Zurich, Zurich 8057
Switzerland

Dear Dr. Gari,

Thank you for submitting your revised manuscript entitled "The iron-sulfur helicase DDX11 promotes the generation of single-stranded DNA for CHK1 signalling". As you will see, while reviewer #2 thinks that it is a pity that the complementation analyses did not work in RPE cells and were not attempted in another cell line, s/he is essentially supportive of publication here now, pending final minor revisions. We would thus be happy to invite you to submit final files to us that address the following:

- Please address the remaining concerns of rev#2 in a point-by-point response and via text changes
- Please provide the supplementary figure files as individual files and without figure legends, the latter should only remain in the main ms file
- For those blots showing mean values and standard deviations from two independent experiments, please instead show the individual values
- Please add author contributions for each author in the author section of the online submission system
- Please provide source data for Fig. 1E

A. FINAL FILES:

B. MANUSCRIPT ORGANIZATION AND FORMATTING:

Sincerely,

Andrea Leibfried, PhD
Executive Editor
Life Science Alliance
Meyerhofstr. 1

69117 Heidelberg, Germany
t +49 6221 8891 502
e a.leibfried@life-science-alliance.org
www.life-science-alliance.org

Reviewer #2 (Comments to the Authors (Required)):

The authors have modified the manuscript according to my comments and the result is satisfactory. It is a pity that the complementation assays did not work in RPE-1 cells and that the authors consider them unfeasible in other cell lines, as I think this would have substantially improved the manuscript. I would recommend the publication of the work, although I still have some minor issues that I hope the authors would be able to address swiftly.

1. Given that the manuscript describes a mechanism that controls the phosphorylation of Chk1 and that no data on other outputs of Chk1 signaling have been added, I think it is more precise to speak of "Chk1 activation" or "Chk1 phosphorylation", particularly in the title of the manuscript and of figures 6 and S4.

2. I appreciate the efforts of the authors to include data on gH2AX on Figure 6. However, I am afraid that the data are not informative since there is no induction of this DNA damage marker as one would expect, for example upon HU and Aph. The authors claim that there is no difference between siDDX11-treated and control cells with respect to gH2AX. With more reason then, they should refer to "Chk1 phosphorylation or activation" and not signaling (see point 1). If Chk1 signaling is affected, then so should gH2AX phosphorylation. I believe these data might confuse the readers, since it is expected that gH2AX augments upon exogenous damage and Chk1 depletion (and therefore DDX1) in an additive fashion. Taking into consideration the data on Attachment 4, I think there might be a general problem with cell culture conditions, since in all cell lines the high levels of gH2AX in control conditions denote high levels of DNA damage, and this should not be the case.

3. Overall, the authors have improved the presentation of the data. They have now included quantifications of western blot experiments. In Figures 6 and S4 they have chosen to show the phospho-Chk1 signals in DDX1-depleted cells as the percentage of those signals in control cells. I believe a more informative way to present the data would be to compare the phospho-Chk1 signal to total Chk1 levels. This kind of analysis, and not the one chosen by the authors, would demonstrate that the decrease in phospho-Chk1 signal upon DDX1 depletion results from reduced phosphorylation and not from reduced Chk1 levels.

4. The authors have considerably improved Figure 5. However, I disagree with their claim that CldU is incorporated at a similar rate in DDX11-depleted and control cells. To make this statement, they should have made the experiment under denaturing conditions. I am sorry if I was not clear enough but my concern is that there is no experiment showing that the phenotypes observed in DDX11-depleted cells are not the consequence of a reduced amount of cycling cells. Maybe this has been proved elsewhere. If not, my concern could have been easily addressed by an EdU or denaturing BrdU IF or FACS. In any case, given the high levels of gH2AX in both control and DDX11-depleted cells, it seems unlikely that DDX11 depletion impacts on the cell cycle profile.

Editor

Thank you for submitting your revised manuscript entitled "The iron-sulfur helicase DDX11 promotes the generation of single-stranded DNA for CHK1 signalling". As you will see, while reviewer #2 thinks that it is a pity that the complementation analyses did not work in RPE cells and were not attempted in another cell line, s/he is essentially supportive of publication here now, pending final minor revisions. We would thus be happy to invite you to submit final files to us that address the following:

- Please address the remaining concerns of rev#2 in a point-by-point response and via text changes
- Please provide the supplementary figure files as individual files and without figure legends, the latter should only remain in the main ms file
- For those blots showing mean values and standard deviations from two independent experiments, please instead show the individual values
- Please add author contributions for each author in the author section of the online submission system
- Please provide source data for Fig. 1E

We have addressed the remaining concerns of Reviewer #2 (see below) and provided the missing files and information as requested.

Reviewer #2 (Comments to the Authors (Required)):

The authors have modified the manuscript according to my comments and the result is satisfactory. It is a pity that the complementation assays did not work in RPE-1 cells and that the authors consider them unfeasible in other cell lines, as I think this would have substantially improved the manuscript. I would recommend the publication of the work, although I still have some minor issues that I hope the authors would be able to address swiftly.

We agree with the reviewer that the complementation analysis would have added to the manuscript and regret that it was not feasible due to technical issues.

1. Given that the manuscript describes a mechanism that controls the phosphorylation of Chk1 and that no data on other outputs of Chk1 signaling have been added, I think it is more precise to speak of "Chk1 activation" or "Chk1 phosphorylation", particularly in the title of the manuscript and of figures 6 and S4.

We agree that it is more correct to speak of CHK1 activation and have changed the title and figure legends accordingly.

2. I appreciate the efforts of the authors to include data on gH2AX on Figure 6. However, I am afraid that the data are not informative since there is no induction of this DNA damage marker as one would expect, for example upon HU and Aph. The authors claim that there is no difference between siDDX11-treated and control cells with respect to gH2AX. With more reason then, they should refer

to "Chk1 phosphorylation or activation" and not signaling (see point 1). If Chk1 signaling is affected, then so should gH2AX phosphorylation. I believe these data might confuse the readers, since it is expected that gH2AX augments upon exogenous damage and Chk1 depletion (and therefore DDX1) in an additive fashion. Taking into consideration the data on Attachment 4, I think there might be a general problem with cell culture conditions, since in all cell lines the high levels of gH2AX in control conditions denote high levels of DNA damage, and this should not be the case.

We are also puzzled by the high pH2AX levels in untreated conditions that we see in all our cell lines (RPE1, HeLa, U2OS). We do not have a good explanation for this, but have found examples of detectable pH2AX levels in untreated conditions also in other papers (e.g. Schmid-JA, Mol Cell, 2018; Haahr-P, Nature Cell Biology, 2016), in which the authors use a number of different cell lines (RPE1, U2OS, HeLa, HC116).

In any case, we agree with the reviewer that these data are rather confusing for the reader. Since we did not observe a difference in pH2AX levels between siC and siDDX11-treated cells, we think it would be best to leave these data out.

3. Overall, the authors have improved the presentation of the data. They have now included quantifications of western blot experiments. In Figures 6 and S4 they have chosen to show the phospho-Chk1 signals in DDX1-depleted cells as the percentage of those signals in control cells. I believe a more informative way to present the data would be to compare the phospho-Chk1 signal to total Chk1 levels. This kind of analysis, and not the one chosen by the authors, would demonstrate that the decrease in phospho-Chk1 signal upon DDX1 depletion results from reduced phosphorylation and not from reduced Chk1 levels.

We have changed the quantifications in Figures 6 and S4 and now show the mean values and standard deviations for % CHK1-pS345/ total CHK1. This data presentation shows very nicely that – although total CHK1 is affected by knock-down of *DDX11* – the levels of CHK1-pS345 per total CHK1 are clearly reduced.

4. The authors have considerably improved Figure 5. However, I disagree with their claim that CldU is incorporated at a similar rate in DDX11-depleted and control cells. To make this statement, they should have made the experiment under denaturing conditions. I am sorry if I was not clear enough but my concern is that there is no experiment showing that the phenotypes observed in DDX11-depleted cells are not the consequence of a reduced amount of cycling cells. Maybe this has been proved elsewhere. If not, my concern could have been easily addressed by an EdU or denaturing BrdU IF or FACS. In any case, given the high levels of gH2AX in both control and DDX11-depleted cells, it seems unlikely that DDX11 depletion impacts on the cell cycle profile.

This was indeed a bit of a misunderstanding. In fact, we had initially tested whether there was an impact of *DDX11* depletion on DNA replication by a QIBC experiment, in which we gave a 20 min EdU pulse. As you can see below, there was no apparent difference in EdU incorporation.

February 6, 2020

RE: Life Science Alliance Manuscript #LSA-2019-00547-TRR

Dr. Kerstin Gari
University of Zurich
Institute of Molecular Cancer Research
Winterthurerstrasse 190
Zurich, Zurich 8057
Switzerland

Dear Dr. Gari,

Thank you for submitting your Research Article entitled "The iron-sulfur helicase DDX11 promotes the generation of single-stranded DNA for CHK1 activation". I appreciate the introduced changes and the further discussion / point-by-point response, and it is a pleasure to let you know that your manuscript is now accepted for publication in Life Science Alliance. Congratulations on this interesting work.

*****IMPORTANT:** If you will be unreachable at any time, please provide us with the email address of an alternate author. Failure to respond to routine queries may lead to unavoidable delays in publication.*******

DISTRIBUTION OF MATERIALS:

Again, congratulations on a very nice paper. I hope you found the review process to be constructive and are pleased with how the manuscript was handled editorially. We look forward to future exciting

submissions from your lab.

Sincerely,
